# THE ROLE OF LEARNING REGIME, ARCHITECTURE AND DATASET STRUCTURE ON SYSTEMATIC GENERALIZATION IN SIMPLE NEURAL NETWORKS

## ABSTRACT

Humans often systematically generalize in situations where standard deep neural networks do not. Empirical studies have shown that the learning procedure and network architecture can influence systematicity in deep networks, but the underlying reasons for this influence remain unclear. Here we theoretically study the acquisition of systematic knowledge by simple neural networks. We introduce a minimal space of datasets with systematic and non-systematic features in both the input and output. For shallow and deep linear networks, we derive learning trajectories for all datasets in this space. The solutions reveal that both shallow and deep networks rely on non-systematic inputs to the same extent throughout learning, such that even with early stopping, no networks learn a fully systematic mapping. Turning to the impact of architecture, we show that modularity improves extraction of systematic structure, but only achieves perfect systematicity in the trivial setting where systematic mappings are fully segregated from non-systematic information. Finally, we analyze iterated learning, a procedure in which generations of networks learn from languages generated by earlier learners. Here we find that networks with output modularity successfully converge over generations to a fully systematic 'language' starting from any dataset in our space. Our results contribute to clarifying the role of learning regime, architecture, and dataset structure in promoting systematic generalization, and provide theoretical support for empirical observations that iterated learning can improve systematicity.

## 1 INTRODUCTION

Humans frequently display the ability to *systematically generalize*, that is, to leverage specific learning experiences in diverse new settings (Lake et al., 2019). For instance, exploiting the approximate compositionality of natural language, humans can combine a finite set of words or phonemes into a near-infinite set of sentences, words, and meanings. Someone who understands "brown dog" and "black cat" also likely understands "brown cat," to take one example from Szabó (2012). The result is that a human's ability to reason about situations or phenomena extends far beyond their ability to directly experience and learn from all such situations or phenomena.

Deep learning techniques have made great strides in tasks like machine translation and language prediction, providing proof of principle that they can succeed in quasi-compositional domains. However, these methods are typically data hungry and the same networks often fail to generalize in even simple settings when training data are scarce (Lake & Baroni, 2018; Lake et al., 2019). Empirically, the degree of systematicity in deep networks is influenced by many factors. One possibility is that the learning dynamics in a deep network could impart an implicit inductive bias toward systematic structure (Hupkes et al., 2020); however, a number of studies have identified situations where depth alone is insufficient for structured generalization (Lake & Baroni, 2018; Niklasson & Sharkey, 1992; Pollack, 1990; Phillips & Wiles, 1993). Another significant factor is architectural modularity, which can enable a system to generalize when modules are appropriately configured (Vani et al., 2021; Phillips, 1995). However, identifying the right modularity through learning remains challenging (Bahdanau et al., 2019). A third possibility builds on Iterated Learning (IL), a method in which generations of agents train briefly on a 'language' produced by their parent, and then generate a new language for their child (Kalish et al., 2007). If systematic components are easier to learn

than non-systematic ones, this process can successively refine a language toward a systematic structure, a process which has been hypothesized to be the cause of the compositional nature of natural language (Kirby, 2001; Kirby et al., 2008). In spite of these (and many other) possibilities for improving systematicity (Hupkes et al., 2020), it remains unclear when standard deep neural networks will exhibit systematic generalization (Dankers et al., 2021), reflecting a long-standing theoretical debate stretching back to the first wave of connectionist deep networks (Rumelhart & McClelland, 1986; Fodor & Pylyshyn, 1988; Smolensky, 1991; 1990; Hadley, 1994).

In this work we theoretically investigate the acquisition of systematic knowledge by simple neural networks (NNs). We introduce a simple space of datasets that contain systematic and non-systematic features, and examine the impact of implicit biases, architectural modularity, and iterated learning on the learned input-output mappings of shallow and deep linear networks. In particular,

- We derive exact training dynamics for shallow and deep linear networks as a function of the dataset parameters.
- We show that for all datasets in the space, despite the possibility of learning a fully systematic mapping, neither shallow nor deep networks do so under gradient descent dynamics.
- We show that modular network architectures can learn fully systematic network mappings, but only when the modularity segregates systematic and non-systematic features.
- We show that iterated learning can converge to a fully systematic 'language' when combined with the weaker architectural constraint of output modularity.

In Section 7 we consider how our findings, which rely on a simplified setting for mathematical tractability, generalize to more complicated datasets and non-linear architectures by training a convolutional neural network to label handwritten digits between 0 and 999. Overall, our results help clarify the diverse factors impacting systematic behavior in neural networks, and suggest that iterated learning can improve systematicity.

## 2 BACKGROUND

Systematic generalization has been proposed as a key feature of intelligent learning agents which can generalize to novel stimuli in their environment (Hockett & Hockett, 1960; Fodor & Pylyshyn, 1988; Hadley, 1993; Kirby et al., 2015; Lake et al., 2017). In particular, the closely related concept of compositional structure has been shown to have benefits for both learning speed (Ren et al., 2019) and generalizability (Lazaridou et al., 2018). There are, however, counter-examples which find only a weak correlation between compositionality and generalization (Andreas, 2018) or learning speed (Kharitonov & Baroni, 2020). In most cases neural networks do not manage to generalize systematically (Ruis et al., 2020), or systematic generalization occurs only with the addition of modular architectures, explicit regularizers or a degree of supervision of the learned features.

Neural Module Networks (NMNs) (Andreas et al., 2016; Hu et al., 2017; 2018) have become one successful method of creating network architectures which generalize systematically. By (jointly) training individual neural modules on particular subsets of data or to perform particular subtasks, the modules will specialize. These modules can then be combined in new ways when an unseen datapoint is input to the model. Thus, through the composition of the modules, the model will systematically generalize, assuming that the correct modules can be structured together. Bahdanau et al. (2019) show, however, that strong regularizers are required for the correct module structures to be learned. Thus, without regularizers, compositional mappings do not emerge with NMNs.

Iterated learning (IL) approaches suggest that systematicity can emerge over generations of agents that learn from each other. In some works, based primarily on the topological structure of language (Brighton & Kirby, 2006), compositional structure does emerge from IL (Ren et al., 2019). When paired with NMNs, IL has previously been used to refine the module structures to be compositional (Vani et al., 2021) by treating the module structure as an output "language". Then each "word" in the language represents a particular module composed to form a "sentence" which describes the model architecture (the relative arrangement of the modules in Polish notation). Thus, by phrasing the problem as a language learning task, IL is able to refine this language of architectures to become modular. This work also highlights the fact that IL is applicable outside of traditional language learning tasks.

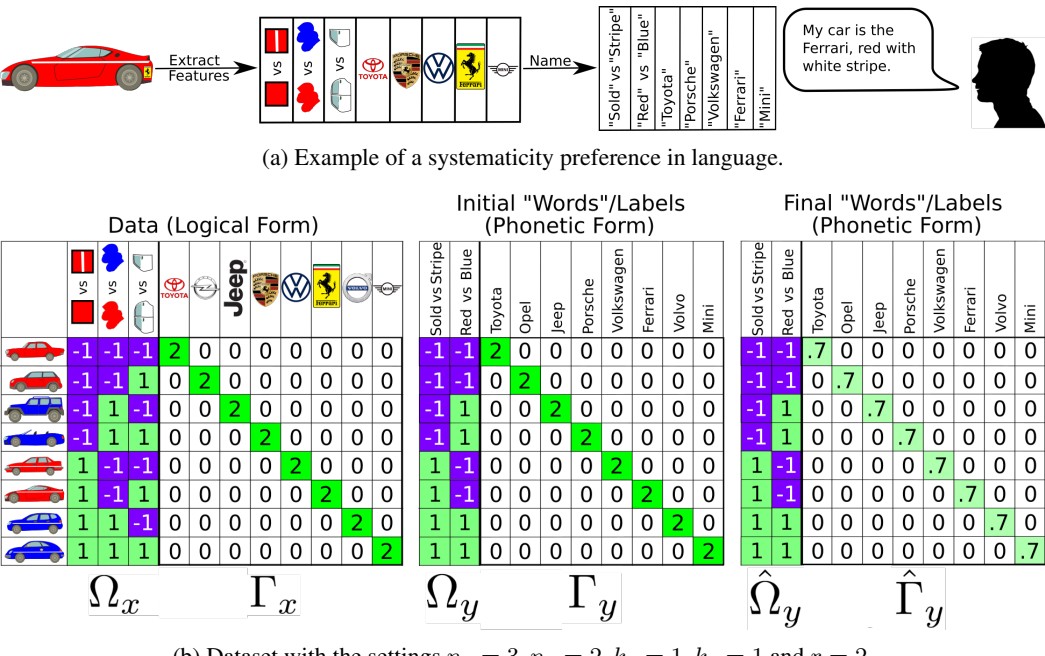

(a) Example of a systematicity preference in language.

(b) Dataset with the settings $n_x = 3, n_y = 2, k_x = 1, k_y = 1$ and $r = 2$.

Figure 1: Problem setting and dataset space. (a) Upon viewing an object, an agent might extract various input features, mapping these to output features or "words" in order to refer to the item (Brighton & Kirby, 2006). (b) We schematize this setting with a space of datasets containing systematic ($\Omega$) and non-systematic ($\Gamma$) features in the input (left panel) and output (middle panel). Rows contain examples and columns contain features. In this case cars are named with both a systematic component (based on features: presence of stripe, colour and number of doors) and non-systematic component (based on brand). In iterated learning, the names of objects change from the initial labels over generations until they stabilize at the refined final labels (right panel, $\hat{\Omega}_y, \hat{\Gamma}_y$).

IL has also been successfully used with NNs as part of a larger pipeline. In particular Seeded Iterated Learning (SIL) (Lu et al., 2020a) uses IL to counter language drift. Language drift occurs when a pretrained language model is then used to complete a language based task. While training on the task, the model begins to lose structure in the learned language that is not useful for performing the task. By training a "teacher" model on task completion and then using the teacher to supervise a pretrained "student", Lu et al. (2020a) show that the student network is able to learn to complete the task while maintaining the initially learned language. SIL is extended by Lu et al. (2020b) to include supervised selfplay in the teacher task-learning step. This is known as Supervised SIL (SSIL). Thus, parts of the data used to pretrain the teacher are also replayed while the teacher learns to perform the task. This also aids in avoiding language drift. While these previous works reflect the influence of certain initial data properties, regularizers or complex architectures on the emergence of compositional structure, the necessity of structured architectures with modular designs remains unclear.

## 3 A SPACE OF DATASETS WITH COMBINATORIAL SYSTEMATICITY

The notion of systematic generalization is broad, and has been assessed using a variety of datasets and paradigms (Hupkes et al., 2020). Here we introduce a simple setting reminiscent of learning to refer to objects in an environment, as depicted in Fig. 1. For instance, we might see a particular car (Fig. 1a), extracting various perceptual input features such as whether it has a stripe, is red, or has a particular hood ornament. This scenario affords several ways in which we could refer to the car. We could systematically map individual input features to individual outputs by saying a sentence like "My car is the red one with the white stripe;" or we could choose a single word to refer to some combination of properties, saying something like "My car is the Ferrari." This task has a type

of combinatorial systematic structure, because individual input features can link independently to individual output features. More generally, an agent first observes an object and forms an internal "logical form". The task is then to map from this logical form to a "phonetic form" by providing a name for each object (Brighton & Kirby, 2006). As illustrated in this example, real world settings often allow redundant expressions and mappings, such that the same object could be individuated with a variety of expressions that vary in systematicity.

To formalize this setting, we define a parametric space of datasets with input and output matrices $X = [\Omega_x \ \Gamma_x]^T$ and $Y = [\Omega_y \ \Gamma_y]^T$ respectively, where $n_x, n_y, k_x, k_y, r \in \mathbb{R}^+$ are the parameters that define a specific dataset. Figure 1b visualizes one dataset in the space. The *systematic input feature* matrix $\Omega_x \in \{-1, 1\}^{n_x \times 2^{n_x}}$ consists of all binary patterns with $n_x$ bits. Here $n_x$ is a key parameter determining the number of bits in the systematic input structure. Overall, the dataset contains $2^{n_x}$ examples. The *systematic output feature* matrix $\Omega_y \in \{-1, 1\}^{n_y \times 2^{n_x}}$ is a sampling of $n_y$ features (rows) from $\Omega_x$, and is the systematic component of the output matrix. Intuitively, the binary combinatorial structure in $\Omega_x$ and $\Omega_y$ is meant to reflect systematic mappings from input features (the color red or blue) to output features (the words "red" or "blue"). Next, the *non-systematic input feature* matrix $\Gamma_x = [rI_1 \ ... \ rI_{k_x}]$ consists of $k_x$ scaled identity matrices, $I_i \in \{0, 1\}^{2^{n_x} \times 2^{n_x}}$. Similarly, the *non-systematic output* matrix $\Gamma_y = [rI_1 \ ... \ rI_{k_y}]$ has $k_y$ scaled identity matrices, with scale factor $r$. These identity matrices provide a single feature for each pattern which is only on for that pattern. Intuitively, these features are meant to reflect idiosyncratic features (like hood ornament, a distinctive scratch, or a proper noun name) or features reflecting specific nonlinear combinations of systematic features (red-striped-2-door). Together $k_x, k_y$ and $r$ control the frequency and intensity (for example the size or perceptual prominence of the brand's logo) of the non-systematic features, which are both factors that can promote non-compositional language being used by humans (Rogers et al., 2004).

This space of datasets is consistent with numerical notions of compositionality in previous works (Andreas, 2018), which define compositionality as a homomorphism between the observation space and the naming space (since the input-output mappings are linear they are homomorphic). The amount of systematic structure can be titrated by adjusting $n_x$ and $n_y$; and the prevalence of non-systematic structure by adjusting $k_x, k_y$, and $r$. Importantly, and novel to our analysis, datasets in this space allow redundant solutions: the systematic output features can be generated based on systematic input features alone, but they can equally be generated using non-systematic features alone, or some mixture of the two. Exploiting this fact, we now ask how reliance on systematic or non-systematic structure is influenced by implicit biases, architecture, and learning regime.

## 4 LEARNING DYNAMICS IN SHALLOW AND DEEP LINEAR NETWORKS

The generalization abilities of deep networks depend on a complex interplay of learning dynamics, architecture, initialization, and dataset statistics. In this section we ask how the implicit inductive bias in gradient descent influences systematicity. We consider training both shallow and deep linear networks on datasets in our space. While deep linear networks can only represent linear input-output mappings, the dynamics of learning change dramatically with the introduction of one or more hidden layers (Fukumizu, 1998; Saxe et al., 2014; 2019; Arora et al., 2018; Lampinen & Ganguli, 2019), and the learning problem becomes non-convex (Baldi & Hornik, 1989). They therefore serve as a tractable model of the influence of depth specifically on learning dynamics, which prior work has shown to impart a low-rank inductive bias on the linear mapping (Huh et al., 2021).

We leverage known exact solutions to the dynamics of learning from small random weights in deep linear networks (Saxe et al., 2014; 2019) to describe the full learning trajectory analytically for every dataset in our space. In particular, consider a single hidden layer network computing output $\hat{y} = W^2 W^1 x$ in response to an input $x$, trained to minimize the mean squared error loss using full batch gradient descent with small learning rate $\epsilon$ (full details and technical assumptions given in Appendix A). The network's total input-output map after $t$ epochs of training is

$$W^2(t)W^1(t) = UA(t)V^T, \tag{1}$$

where $A(t)$ is a diagonal matrix of singular values. The dynamics of $A(t)$, as well as the orthogonal matrices $U$ and $V$, depend on the singular value decomposition of the input- and input-output correlations in the dataset. If the input- and input-output correlations can be expressed as

$$\Sigma^x = E[XX^T] = VDV^T, \quad \Sigma^{yx} = E[YX^T] = USV^T \tag{2}$$

where $U$ and $V$ are orthogonal matrices of singular vectors and $S, D$ are diagonal matrices of singular values/eigenvalues, then the diagonal elements of $A(t)_{\alpha\alpha} = \pi_\alpha(t)$ evolve through time as

$$\pi_\alpha(t) = \frac{\lambda_\alpha/\delta_\alpha}{1 - \left(1 - \frac{\lambda_\alpha}{\delta_\alpha \pi_0}\right)\exp\left(\frac{-2\lambda_\alpha}{\tau}t\right)}, \tag{3}$$

where $\lambda_\alpha$ and $\delta_\alpha$ are the associated input-output singular value and input eigenvalue ($S_{\alpha\alpha}$ and $D_{\alpha\alpha}$ respectively), $\pi_0$ denotes the singular value at initialization, and $\tau = \frac{1}{2^{n_x}\epsilon}$ is the learning time constant. These dynamics describe a trajectory which begins at the initial value $\pi_0$ when $t = 0$ and increases to $\lambda_\alpha/\delta_\alpha$ as $t \to \infty$.

In essence, the network's total input-output mapping at all times in training is a function of the singular value decomposition of the dataset statistics. We therefore analytically obtained this decomposition in terms of the dataset parameters for our space of datasets. We find that there are three distinct input-output singular values which we denote $\lambda_1$, $\lambda_2$ and $\lambda_3$; two distinct input singular values $\delta_1$ and $\delta_2$; and therefore three asymptotes $\pi_1^{ss}, \pi_2^{ss}$ and $\pi_3^{ss}$,

$$\lambda_1 = \left(\frac{(k_x r^2 + 2^{n_x})(k_y r^2 + 2^{n_x})}{2^{2n_x}}\right)^{\frac{1}{2}} \quad (4) \qquad \pi_1^{ss} = \lambda_1/\delta_1 = \left(\frac{k_y r^2 + 2^{n_x}}{k_x r^2 + 2^{n_x}}\right)^{\frac{1}{2}} \quad (5)$$

$$\lambda_2 = \left(\frac{(k_x r^2 + 2^{n_x})(k_y r^2)}{2^{2n_x}}\right)^{\frac{1}{2}} \quad (6) \qquad \pi_2^{ss} = \lambda_2/\delta_1 = \left(\frac{k_y r^2}{k_x r^2 + 2^{n_x}}\right)^{\frac{1}{2}} \quad (7)$$

$$\lambda_3 = \left(\frac{k_x k_y r^4}{2^{2n_x}}\right)^{\frac{1}{2}} \quad (8) \qquad \pi_3^{ss} = \lambda_3/\delta_2 = \left(\frac{k_y}{k_x}\right)^{\frac{1}{2}} \quad (9)$$

Substituting these expressions into the dynamics yields the full learning trajectories for all datasets in the space. We note that each distinct singular value occurs multiple times in the dataset: $\lambda_1$ has multiplicity $n_x - n_y$, $\lambda_2$ has multiplicity $n_y$, and $\lambda_3$ has multiplicity $2^{n_x} - n_x$. Full derivations, including explicit expressions for singular vectors, are deferred to Appendix B.

A similar derivation for a shallow network (no hidden layer) shows that the singular values of the model's mapping follow the trajectory

$$\pi_\alpha(t) = \lambda_\alpha/\delta_\alpha\left(1 - \exp\left(-\delta_\alpha t/\tau\right)\right) + \pi_0 \exp\left(-\delta_\alpha t/\tau\right), \tag{10}$$

such that the time course depends on the singular values of the input covariance matrix, $\Sigma^x$. The unique singular values are

$$\delta_1 = \frac{(k_x r^2 + 2^{n_x})}{2^{n_x}} \quad (11) \qquad \delta_2 = \frac{k_x r^2}{2^{n_x}}. \quad (12)$$

We empirically verify these equations by simulating the full training dynamics for deep and shallow linear networks trained using gradient descent on an instantiation from the space of datasets in Figure 2. While training, we compute the singular values of the network after each epoch of training. These simulations of the training dynamics for each unique singular value are then compared to the predicted dynamics. We see close agreement between the predicted and simulated trajectories.[1]

### 4.1 THE EVOLUTION OF SYSTEMATICITY OVER LEARNING

To understand the extent to which a network comes to rely on systematic or non-systematic input features, and the timing with which it produces systematic or non-systematic output features, we calculate the Frobenius norm of the input-output mapping between different subsets of features. In particular we partition the input-output mapping into four components: systematic inputs to systematic outputs; non-systematic inputs to systematic outputs; systematic inputs to non-systematic outputs; and non-systematic outputs to non-systematic outputs. Analytical expressions for these norms over training (which rely on both singular value dynamics and the structure of the singular

---

[1] All experiments are run using the Jax library (Bradbury et al., 2018). Full code for reproducing all figures is contained in the Supplementary Material.

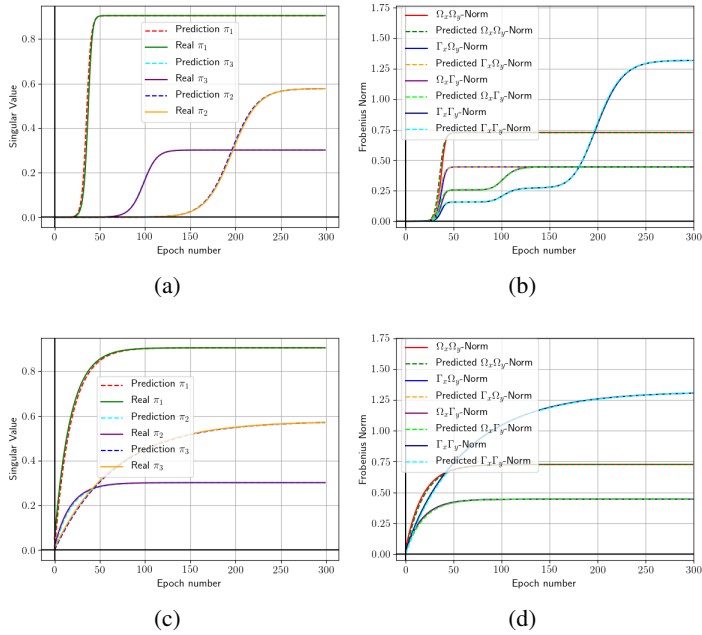

Figure 2: Analytical learning dynamics for deep (panels a-b) and shallow (panels c-d) linear networks. (a,c) Comparison of predicted and actual singular value trajectories over learning, for the three unique dataset singular values. (b,d) Comparison of predicted and actual Frobenius norms of the input-output mapping to/from systematic ($\Omega_x, \Omega_y$) and non-systematic ($\Gamma_x, \Gamma_y$) features. Deep networks show distinct stages of improvement over learning. However, at no point is a purely systematic mapping learned. *Parameters*: $n_x = 3, n_y = 1, k_x = 3, k_y = 1, r = 1$.

vectors) are given in Appendix D Eqns. 17-20 due to space constraints. Figure 2b,c depicts these dynamics for one specific dataset. Crucially, we find that, first, systematic outputs are generated using substantial contributions from non-systematic inputs, regardless of network depth (see $\Gamma_x\Omega_y$ curve); and second, contributions to non-systematic outputs $\Gamma_y$ arise simultaneously with contributions to systematic outputs $\Omega_y$ (all curves initially rise together). Hence perfectly systematic mappings never arise at any point in the learning process, either for shallow or deep networks. For these datasets in which a given mapping can be implemented using either systematic or non-systematic features, the implicit bias in learning dynamics settles on a mixed solution.

While the results in Figure 2 are for one specific dataset, our analytical results offer more general insight throughout our space of datasets. We note that in the deep network, the timescale of learning each unique singular value is roughly $O(1/\lambda)$, such that larger singular values are learned faster (Saxe et al., 2019). In Appendix A we show that $\lambda_1 > \lambda_2 > \lambda_3$ for any setting of the dataset parameters, and $\pi_1^{ss} \geq \pi_3^{ss} > \pi_2^{ss}$ assuming that $k_x \geq k_y$ (i.e., the non-systematic output dimension is not larger than the non-systematic input dimension). Further, the three classes of singular vectors associated with each singular value occur due to specific properties of the dataset. $\lambda_1$, and by extension $\pi_1$, arises from the systematic input and output features, and it makes contributions to all four norms. $\lambda_2$ occurs due to the left-out systematic output features, and contributes only to the norms of the mappings to non-systematic outputs. Lastly, $\lambda_3$ occurs because of the non-systematic features, and appears only in the norm from non-systematic inputs to non-systematic outputs. Thus, for all datasets in the space, the mixed systematic/non-systematic singular value mode $\pi_1$ is the fastest singular value to learn, while the purely non-systematic singular value $\pi_3$ is the second largest but slowest to learn by the network.

In the shallow network, $\delta_1$ drives the dynamics of the mixed systematic/non-systematic mode and $\delta_2$ contributes to the non-systematic mappings. We can show that for any setting of the dataset parameters $\delta_1 > \delta_2$ and so the mixed systematic/non-systematic mapping is again learned faster

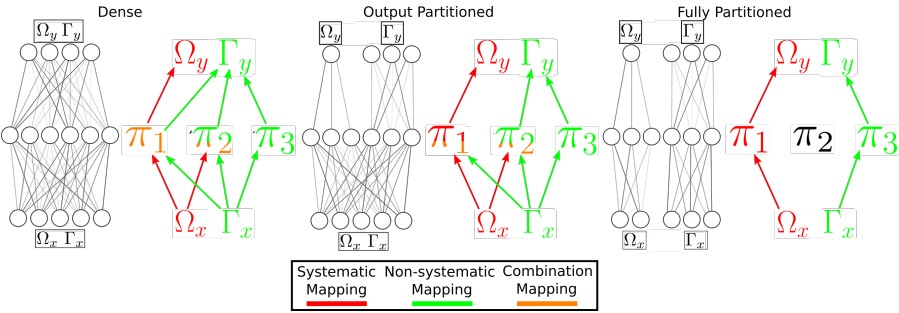

Figure 3: Impact of architectural biases. Architectures that partition systematic and non-systematic features in different ways with the corresponding graphical representation of the resulting network mappings. The dynamical modes $\pi_1$, $\pi_2$, and $\pi_3$ contain contributions from systematic and non-systematic input components, depicted as the color of the bottom half of the each mode; and they make contributions to systematic and non-systematic output features, depicted as the color of the top half of each mode (red for systematic, green for non-systematic, and orange for mixed). To learn a systematic mapping, the fastest mode $\pi_1$ must be systematic. The output partitioned network is able to obtain output systematicity but not input systematicity. Only the fully partitioned network obtains complete systematicity.

than the purely non-systematic one. However, because the shallow dynamics exhibit exponential approach to their asymptote, both modes are learned at overlapping times.

In sum, a part of the non-systematic mapping will be learned at the same speed as the systematic mapping, for any setting of our dataset. Thus, there is no epoch at which training could be stopped that would completely remove mappings from the non-systematic component of the input and output. In our setting, the implicit bias arising from depth, small random initialization, and gradient descent is insufficient to learn fully systematic mappings for any datasets in this space.

## 5 MODULARITY AND NETWORK ARCHITECTURE

We now turn to modularity and network architecture, another prominent approach for promoting systematicity in a network's mapping (Vani et al., 2021; Bahdanau et al., 2019). Architectures such as Neural Module Networks (Andreas et al., 2016; Hu et al., 2017; 2018) learn reconfigurable modules that implement specific aspects of a larger problem. By rearranging existing modules to process a novel input, they can generalize far beyond their training set. Here we investigate whether simple forms of additional architectural structure can yield strong enough inductive biases to learn fully systematic mappings.

In particular, instead of a dense network, we consider architectures in which systematic and non-systematic features are processed in separate processing streams, as depicted in Figure 3. The learning dynamics for several of these different settings can be obtained by combining the dynamics of other points in the space of datasets. To illustrate the approach, consider the output-partitioned network. The pathway leading to the systematic outputs effectively learns from a dataset with $k_y = 0$, and the pathway leading to the non-systematic outputs effectively learns from a dataset with $n_y = 0$. By combining these dynamics, we can reconstruct the full solution, as shown in Figure 8 of Appendix E. We note that this approach cannot be taken for the input partitioned network, which has interactions in the output layer. We summarize our findings with the graphical representations of the network mappings in Figure 3. The same three modes drive learning dynamics, but they contain subtly different mixtures of contributions from systematic and non-systematic inputs and outputs. Notably, the output-partitioned network exhibits output systematicity but not input systematicity, a significant point that we will return to in the iterated learning section. However, only the fully partitioned network achieves full systematicity, such that early stopping could completely prune non-systematic features. Hence we find that architectural biases can enforce systematicity, but only in the heavy-handed and relatively trivial case where the bias perfectly segregates systematic and non-systematic information.

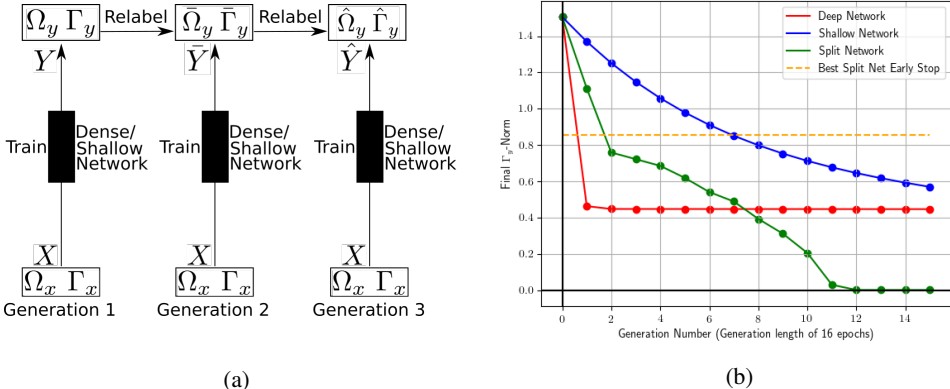

(a)  (b)

Figure 4: Iterated learning dynamics. (a) Generations of agents learn from languages generated by their parent, and pass on their acquired language to their children. (b) Norm of non-systematic output mapping over 20 generations of the IL procedure using deep, shallow and output-split networks. Dashed line shows the best early-stopping can achieve for the starting dataset (stopping at epoch 30 with the output-split network). Even though the networks are run for 40 epochs per generation, IL is still able to out-perform early stopping with the output-split network. It is important to note that the dense and split networks maintain the original systematic mapping by epoch 40, and these results show the closest each network can get to removing the non-systematic component under this restriction. The shallow network is unable to fully maintain the systematic mapping with any form of IL, thus the decrease in its norm is at the expense of the systematic mapping. *Parameters*: $n_x = 3, n_y = 2, k_x = 1, k_y = 1$ and $r = 2$.

## 6 ITERATED LEARNING

Finally, we consider the impact of iterated learning on extracting systematic structure, alongside architectural partitioning. In iterated learning, the focus is on refining the output 'language' toward systematicity over generations of learners (Kirby, 2001; Kalish et al., 2007), as illustrated in Figure 1b middle and right panel. Each generation learns from the language acquired by the previous generation (Figure 4a). To instantiate this setting, we start from a particular dataset in our space, but halt training before full convergence after a pre-defined number of training steps. We then use the network's output (logits) as the target outputs for the next generation. Throughout learning, the network's input-output mapping takes the same form as the dataset's singular value decomposition, but with different singular values. Therefore, the generated language always has the same singular vectors, but with different singular values corresponding to the amount of learning progress made by a given agent. This fact permits straightforward analysis of iterated learning dynamics throughout our space of datasets.

Figure 4b depicts the refinement of the output language over generations in networks with different architectures. In particular, iterated learning is effective in reducing or eliminating the non-systematic output features, and can do so more effectively than carefully chosen early-stopping on the original dataset. We make several observations. First, while deep and shallow networks do not learn perfectly systematic mappings in the first generation, the deep network does learn a relatively more systematic mapping, such that IL yields more rapid benefits. Second, iterated learning in dense networks can reduce non-systematic structure relative to early stopping, but does not eliminate it. Third, and most notably, iterated learning can exploit weak architectural priors: with optimal early stopping, the output partitioned network does not fully remove non-systematic output features, but over generations of iterated learning it completely eliminates non-systematic output structure.

This effect arises because the systematic and non-systematic mappings depend on entirely different modes. Since $\pi_1$ is used by the systematic mapping and is learned the fastest for the space of datasets, in fact IL in the output-partitioned network completely removes the non-systematic mapping for any dataset in the space of datasets. Whereas early-stopping has varying performance over the space of datasets, IL with the output-split network will always be able to converge to a systematic language

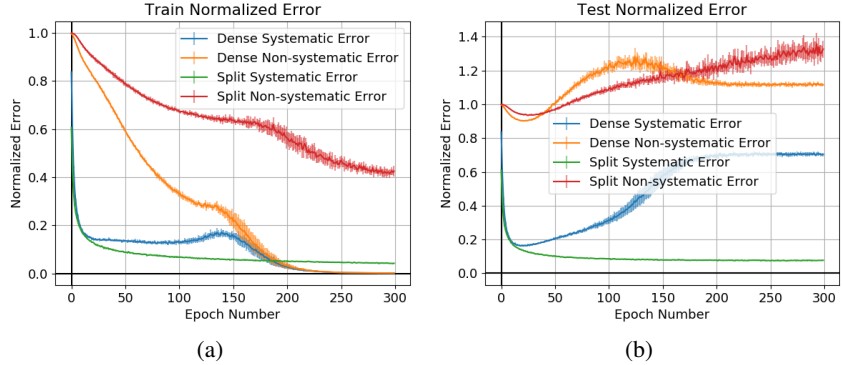

Figure 5: (a) Normalized training loss, (b) test loss of a deep CNN with ReLU activation on the Compositional-MNIST dataset averaged over 10 runs. Error bars reflects one standard-deviation.

(dataset labels). However, we note that still, this language will be produced in part from mappings from non-systematic input features. Thus IL only achieves output systematicity.

# 7 COMPOSITIONAL MNIST (CMNIST)

To evaluate how well our results generalize to non-linear networks and more complex datasets, in this section we train a deep Convolutional Neural Network (CNN) to learn a compositional variant of MNIST (CMNIST). In this dataset three digits from MNIST are stacked horizontally, resulting in a value between 0 and 999. The systematic output encodes each digit in a 10-way one-hot vector, resulting in a 30-dimensional vector. The non-systematic output encodes the number as a whole with a 1000-dimensional one-hot vector. This task is similar to the SVHN dataset (Netzer et al., 2011) with systematic and non-systematic output labels.

We compare results of using a single dense CNN and a split CNN, in which two parallel sets of convolutional layers with half the convolutional filters of the dense network each that connect separately to systematic and non-systematic labels. Full details of the two network architectures and hyper-parameters used for the CMNIST experiments are given in Appendix G. The effect predicted by our theoretical work is that the redundancy in the labels will interfere with the network's learned hidden representations and decrease the systematic generalizability of the network. This prediction is shown to be true in Figure 5, which shows the mean-squared error for the systematic and non-systematic network predictions over the course of training, normalized so that the initial error is at 1.0. Firstly, in dense networks the error of the systematic mapping is tied to the error of the non-systematic mapping. This is seen in Figure 5(a) where the blue curve cannot converge until the orange curve has become sufficiently low. This effect is not observed with the split network. Secondly, comparing Figure 5(a) and 5(b) we demonstrate the lack of generalization which occurs when using non-systematic features. This is seen as the orange and red curves achieving a lower training error while the test error increases. Lastly, again by comparing Figure 5(a) and 5(b), we see that the systematic mapping of the split architecture is the only mapping which generalizes well (near 0 training and test error). Thus, even in this more complex setting, we see the negative effect a shared hidden layer has on the generalization of the network (comparing the blue and green curves).

# 8 DISCUSSION

In this work we have theoretically and empirically studied the ability of simple NNs to acquire systematic knowledge. We found that this ability is challenging even in our simple setting. Neither implicit biases in learning dynamics, nor all but the most stringent modularity, caused networks to ignore non-systematic inputs. However, iterated learning–when combined with weaker architectural priors–can converge to strongly systematic output features. Our results complement recent empirical studies, helping to highlight the complex factors influencing systematic generalization. We hope greater understanding of these phenomena will ultimately aid the design of improved learning systems that can leverage specific learning in diverse new ways.

## 9 REPRODUCIBILITY STATEMENT

To aid the reproducibility of this work, we have included all source code used to obtain the results in the supplementary material as well as instructions on using the code. In this work we introduced the space of datasets used in Sections 4 and 5, for our theoretical analysis with linear networks, as well as the Compositional MNIST dataset. Included in the source code are methods of constructing the space of datasets and for loading the CMNIST dataset. No pre-processing steps are used on either dataset before being used to train the networks. In Appendix G we have also included all hyper-parameters and network architectures used for the CMNIST experiments. Finally, for our theoretical results, the full details and technical assumptions can be found in Appendix A.

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

# A    LEARNING DYNAMICS IN SHALLOW AND DEEP LINEAR NETWORKS

The dynamics of learning for shallow and deep linear networks are derived in Saxe et al. (2019). We state full details of our specific setting here. We train a linear network with one hidden layer to minimize the quadratic loss $L(W^1, W^2) = \frac{1}{2^{n_x}}||Y - W^2 W^1 X||_2^2$ using gradient descent. This gives the learning rules $E[\Delta W^1] = \frac{\epsilon}{2^{n_x}} W^{2^T}(Y - W^2 W^1 X) X^T$ and $E[\Delta W^2] = \frac{\epsilon}{2^{n_x}}(Y - W^2 W^1 X)(W^1 X)^T$. By using a small learning rate $\epsilon$ and taking the continuous time limit, the mean change in weights is given by $\tau \frac{d}{dt} W^1 = W^{2^T}(\Sigma^{yx} - W^2 W^1 \Sigma^x)$ and $\tau \frac{d}{dt} W^2 = (\Sigma^{yx} - W^2 W^1 \Sigma^x) W^{1^T}$ where $\Sigma^x = E[XX^T]$ is the input correlation matrix, $\Sigma^{yx} = E[YX^T]$ is the input-output correlation matrix and $\tau = \frac{1}{2^{n_x}\epsilon}$. Here, $t$ measures units of learning epochs. It is helpful to note that since we are using a small learning rate the full batch gradient descent and stochastic gradient descent dynamics will be the same. Saxe et al. (2019) has shown that the learning dynamics depend on the singular value decomposition of $\Sigma^{yx} = USV^T = \sum_{\alpha=1}^{min(n_x+k_x 2^{n_x}, n_y+k_y 2^{n_x})} \lambda_\alpha u^\alpha v^{\alpha^T}$ and $\Sigma^x = VDV^T = \sum_{\alpha=1}^{n_x+k_x 2^{n_x}} \delta_\alpha u^\alpha v^{\alpha^T}$. To solve for the dynamics we require that the right singular vectors $V$ of $\Sigma^{yx}$ are also the singular vectors of $\Sigma^x$. This is the case for any dataset in our space, as shown in Appendix B. Note, we assume that the network has at least $2^{n_x}$ hidden neurons (the number of singular values in the input-output covariance matrix) so that it can learn the desired mapping perfectly. If this is not the case then the model will learn the top $n_h$ singular values of the input-output mapping where $n_h$ is the number of hidden neurons (Saxe et al., 2014). Given the SVDs of the two correlation matrices the learning dynamics can be described explicitly as $W^2(t)W^1(t) = UA(t)V^T = \sum_{\alpha=1}^{min(n_x+k_x 2^{n_x}, n_y+k_y 2^{n_x})} \pi_\alpha(t) u^\alpha v^{\alpha T}$ where $A(t)$ is the effective singular value matrix of the network's mapping. The trajectory of each singular value in $A(t)$ is described as $\pi_\alpha(t) = \frac{\lambda_\alpha/\delta_\alpha}{1-(1-\frac{\lambda_\alpha}{\delta_\alpha \pi_0})\exp(\frac{-2\lambda_\alpha}{\tau}t)}$. From these dynamics it is helpful to note that the time-course of the trajectory is only dependent on the $\Sigma^{yx}$ singular values. Thus, $\Sigma^x$ affects the stable point of the network singular values but not the time-course of learning. In addition for the $\Sigma^{yx}$ singular values we have Theorem 1:

**Theorem 1** *For all points in the space of datasets: $n_x, n_y, k_x, k_y, r \in \mathbb{R}^+$ the input-output covariance matrix $\Sigma^{yx}$ singular values will be ordered as: $\lambda_1 > \lambda_2 > \lambda_3$.*

**Proof:**
Firstly we prove that $\lambda_1 > \lambda_2$:

$$\lambda_1 > \lambda_2$$
$$\left(\frac{(k_x r^2 + 2^{n_x})(k_y r^2 + 2^{n_x})}{2^{2n_x}}\right)^{\frac{1}{2}} > \left(\frac{(k_x r^2 + 2^{n_x})(k_y r^2)}{2^{2n_x}}\right)^{\frac{1}{2}}$$
$$\left((k_x r^2 + 2^{n_x})(k_y r^2 + 2^{n_x})\right) > \left((k_x r^2 + 2^{n_x})(k_y r^2)\right)$$
$$k_y r^2 + 2^{n_x} > k_y r^2$$
$$2^{n_x} > 0$$

$2^{n_x} > 0$ is true by definition since $n_x \in \mathbb{R}^+$ and, thus, $\lambda_1 > \lambda_2$ for all points in our space of datasets.

Now we prove that $\lambda_2 > \lambda_3$:

$$\lambda_2 > \lambda_3$$
$$\left(\frac{(k_x r^2 + 2^{n_x})(k_y r^2)}{2^{2n_x}}\right)^{\frac{1}{2}} > \left(\frac{k_x k_y r^4}{2^{2n_x}}\right)^{\frac{1}{2}}$$
$$(k_x r^2 + 2^{n_x})(k_y r^2) > k_x k_y r^4$$
$$k_x k_y r^4 + 2^{n_x} k_y r^2 > k_x k_y r^4$$
$$2^{n_x} k_y r^2 > 0$$

$2^{n_x} k_y r^2 > 0$ is true by definition since $n_x, k_x, r \in \mathbb{R}^+$ and, thus, $\lambda_2 > \lambda_3$ for all points in our space of datasets. Thus, using the transitivity of inequality: $\lambda_1 > \lambda_2 > \lambda_3$ for all points in the space of datasets.

## B  SINGULAR VALUE DECOMPOSITION EQUATIONS

In this section we provide the general formulas for the singular value decomposition of the $\Sigma^x$ and $\Sigma^{yx}$ covariance matrices for any dataset in the space of datasets. As stated in Section 4 the right singular vectors of $\Sigma^{yx}$ must match the singular vectors of $\Sigma^x$ which is the $V$ matrix below. Thus, $\Sigma^x = VDV^T$ and $\Sigma^{yx} = USV^T$.

Let:
$A = (\Omega_y \Omega_x^T)^T \Omega_y \Omega_x^T$
$B = \Omega_y^T \Omega_y \Omega_x^T$
$C = \Omega_x^T \Omega_x \Omega_x^T$
$THP^T = (\frac{1}{k_x k_y})^{\frac{1}{4}} \mathbf{I_{2^{n_x} \times 2^{n_x}}} - (\frac{1}{k_x k_y})^{\frac{1}{4}} (\frac{1}{2^{n_x}}) \Omega_x^T \Omega_x$
Where $THP^T$ is the SVD of $(\frac{1}{k_x k_y})^{\frac{1}{4}} \mathbf{I_{2^{n_x} \times 2^{n_x}}} - (\frac{1}{k_x k_y})^{\frac{1}{4}} (\frac{1}{2^{n_x}}) \Omega_x^T \Omega_x$. Then the following are the matrix formulas for the components of the SVD for $\Sigma^{yx}$ and $\Sigma^x$.

$$U = \begin{bmatrix} \left(\frac{1}{2^{n_x}(k_y r^2 + 2^{n_x})}\right)^{\frac{1}{2}} \Omega_y \Omega_x^T & \mathbf{0_{n_y \times k_x 2^{n_x}}} \\ \left(\frac{r^2}{2^{3n_x}(k_y r^2 + 2^{n_x})}\right)^{\frac{1}{2}} B + \left(\frac{r^2}{2^{3n_x}(k_y r^2)}\right)^{\frac{1}{2}} (C - B) & T \end{bmatrix} \tag{13}$$

$$V^T = \begin{bmatrix} \left(\frac{2^{n_x}}{(k_x r^2 + 2^{n_x})}\right)^{\frac{1}{2}} \mathbf{I_{n_x \times n_x}} & \left(\frac{r^2}{2^{n_x}(k_x r^2 + 2^{n_x})}\right)^{\frac{1}{2}} \Omega_x \\ \mathbf{0_{k_x 2^{n_x} \times n_x}} & (\frac{1}{k_x})^{\frac{1}{2}} P^T \end{bmatrix} \tag{14}$$

$$S = \begin{bmatrix} \left(\frac{(k_x r^2 + 2^{n_x})(k_y r^2 + 2^{n_x})}{2^{6n_x}}\right)^{\frac{1}{2}} A + \left(\frac{(k_x r^2 + 2^{n_x})(k_y r^2)}{2^{2n_x}}\right)^{\frac{1}{2}} (I - \frac{1}{2}A) & \mathbf{0_{n_x \times k_x 2^{n_x}}} \\ \mathbf{0_{n_x \times k_x 2^{n_x}}} & (k_x k_y)^{\frac{1}{2}} \frac{r^2}{2^{n_x}} \mathbf{I_{2^{n_x} \times 2^{n_x}}} \end{bmatrix} \tag{15}$$

$$D = \begin{bmatrix} \left(\frac{(k_x r^2 + 2^{n_x})}{2^{n_x}}\right)^{\frac{1}{2}} \mathbf{I_{n_x \times n_x}} & \mathbf{0_{n_x \times k_x 2^{n_x}}} \\ \mathbf{0_{k_x 2^{n_x} \times n_x}} & \frac{k_x r^2}{2^{n_x}} \mathbf{I_{2^{n_x} \times 2^{n_x}}} \end{bmatrix} \tag{16}$$

## C  SHALLOW NETWORK SIMULATIONS

In this section we show the simulations of a shallow network trained on an instance from the space of datasets. We again see strong agreement between the predicted and simulated time-courses for the training dynamics. Importantly for our findings in Section 4, the modes of variation for a shallow network are all learned at roughly the same time. This can be seen in Figure 6 where the time-courses, and by extension norms, are all learned simultaneously. Thus, adding depth to a network is necessary for IL to be even partially effective at removing non-systematic output labels while maintaining the systematic outputs.

## D  INPUT AND OUTPUT PARTITIONED FROBENIUS NORMS

We partition the input-output mapping along the systematic and non-systematic input and output components. The time-courses for these norms can be seen in Equations 17 to 20. From these equations we see that the mappings to both components of the output rely on all inputs. Thus, the non-systematic inputs still offer some benefit to the systematic output and are used in the systematic mapping. Likewise the systematic inputs are used for the non-systematic mapping.

$$\Omega_x\text{-}\Omega_y\text{-Norm} = \left(\frac{2^{2n_x} n_y \pi_1^2(t)}{(k_x r^2 + 2^{n_x})(k_y r^2 + 2^{n_x})}\right)^{\frac{1}{2}} \tag{17}$$

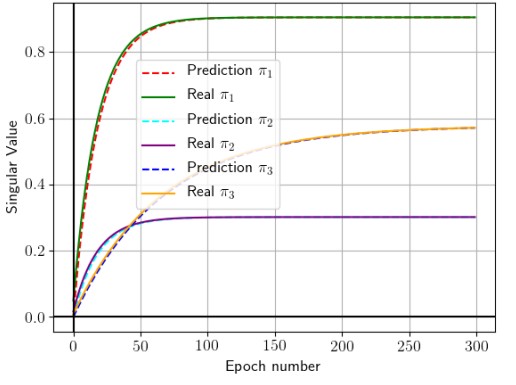
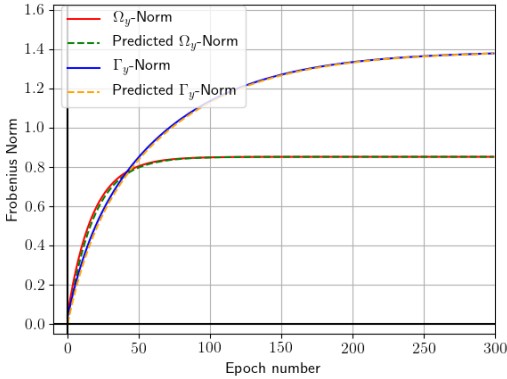

(a) Learning trajectories of the three unique Singular Values learned by the shallow network.

(b) Time paths of the shallow network mappings

Figure 6: Singular Values and Frobenius norms of the shallow network mappings

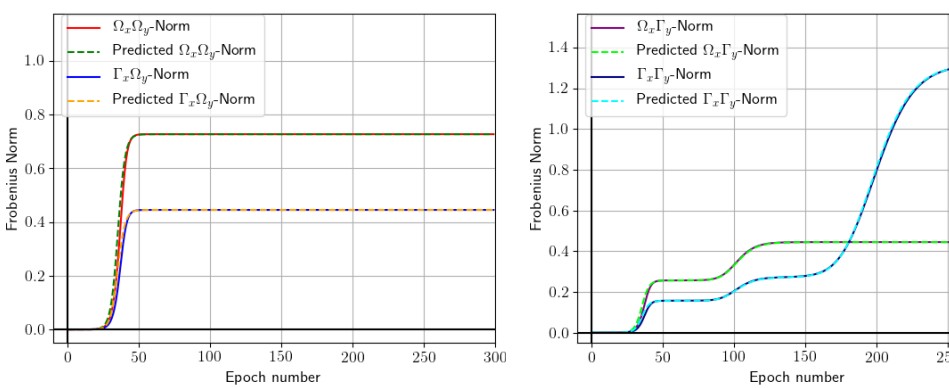

(a) Systematic output norm partitioned by the systematic and non-systematic input components.

(b) Non-systematic output norm partitioned by systematic and non-systematic input components.

Figure 7: Frobenius Norm of the systematic and non-systematic deep network mapping partitioned by the systematic and non-systematic inputs on a dense linear network.

$$\Gamma_x\text{-}\Omega_y\text{-Norm} = \left( \frac{2^{n_x} n_y k_x r^2 \pi_1^2(t)}{(k_x r^2 + 2^{n_x})(k_y r^2 + 2^{n_x})} \right)^{\frac{1}{2}} \tag{18}$$

$$\Omega_x\text{-}\Gamma_y\text{-Norm} = \left( \frac{2^{n_x} k_y n_y r^2 \pi_1^2(t)}{(k_x r^2 + 2^{n_x})(k_y r^2 + 2^{n_x})} + \frac{2^{n_x}(n_x - n_y)}{k_x r^2 + 2^{n_x}} \pi_2^2(t) \right)^{\frac{1}{2}} \tag{19}$$

$$\Gamma_x\text{-}\Gamma_y\text{-Norm} = \left( \frac{k_x k_y n_y r^4 \pi_1^2(t)}{(k_x r^2 + 2^{n_x})(k_y r^2 + 2^{n_x})} + \frac{(n_x - n_y)k_x r^2}{k_x r^2 + 2^{n_x}} \pi_2^2(t) + (2^{n_x} - n_x)\pi_3^2(t) \right)^{\frac{1}{2}} \tag{20}$$

Figure 7 shows the simulated and predicted training dynamics for these norms on a deep linear network. Figure 7a reflects that the mapping to the systematic output relies evenly on both the systematic and non-systematic inputs, with a slight preference to the systematic inputs. Likewise, Figure 7b reflects that the mapping to the non-systematic outputs also uses the systematic and non-systematic inputs evenly, with a preference towards the non-systematic inputs towards the end of training.

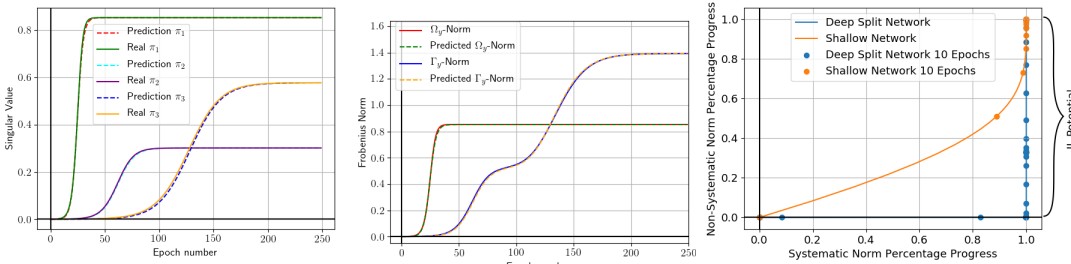

(a) Learning trajectories of the three unique Singular Values learned by the output-split network.

(b) Time paths of the split network mappings

(c) Phase diagram of the split network vs shallow network Frobenius norms

Figure 8: Singular Values and Frobenius norms of the split network mappings

We can also consider the Frobenius norm from the full input to either the systematic or non-systematic components of the output (obtained by summing over input features). This takes a simpler form,

$$\Omega_y\text{-Norm} = \left(\frac{2^{n_x} n_y \pi_1^2(t)}{k_y r^2 + 2^{n_x}}\right)^{\frac{1}{2}} \tag{21}$$

$$\Gamma_y\text{-Norm} = \left(\frac{k_y n_y r^2 \pi_1^2(t)}{k_y r^2 + 2^{n_x}} + (n_x - n_y)\pi_2^2(t) + (2^{n_x} - n_x)\pi_3^2(t)\right)^{\frac{1}{2}}. \tag{22}$$

## E  MODULARITY AND ARCHITECTURE

In Figure 8 we depict the Frobenius norm of the mapping from the entire input to the systematic and non-systematic output, for the output-partitioned network. The dynamics of this case can equally be seen as using one neural module responsible for the systematic mapping and another for the non-systematic mapping. Now, from the perspective of the non-systematic mapping, none of the systematic features in the input are present in the output, and the mapping will no longer be dependent on $\pi_1$. The systematic mapping is still only dependent on the $\pi_1$ effective singular value. The time-courses for the resultant Frobenius norms are given in Equations 23 and 24.

$$\Omega_y\text{-Norm} = \left(n_y \pi_1^2(t)\right)^{\frac{1}{2}} \tag{23}$$

$$\Gamma_y\text{-Norm} = \left(n_x \pi_2^2(t) + (2^{n_x} - n_x)\pi_3^2(t)\right)^{\frac{1}{2}} \tag{24}$$

## F  IMPERFECT OUTPUT PARTITIONS

We now investigate the case where the split network architecture does not perfectly partition the output into the systematic and non-systematic components. In this case some of the non-systematic identity output blocks are grouped with the systematic outputs (we only consider partitioning along the systematic and non-systematic blocks to keep the closed form solutions tractable). Thus, we separate the number of non-systematic outputs $k_y$ into the number of non-systematic outputs of the left network branch $k_y^{\text{left}}$ and right network branch $k_y^{\text{right}}$, such that $k_y^{\text{left}} + k_y^{\text{right}} = k_y$. We also examine the mapping from the full input to aid in readability.

With this setup three distinct Frobenius norms emerge, as shown in Equations 25, In the extreme cases when $k_y^{\text{left}} = k_y$ we recover the dense network equations shown in Equations 21 and 22, since $\pi_2 = \pi_3 = 0$ for Equation 27. This is apparent from the singular value equations for $\pi_2$ and $\pi_3$ shown in Equations 7 and 9 with $k_y$ in the numerator which is replaced by $k_y^{\text{right}}$ in this case. Thus,

Equation 27 falls away completely. In the other extreme case of $k_y^{\text{right}} = k_y$ we recover the split network equations shown in Equations 23 and 24. This is again because $\pi_2 = \pi_3 = 0$ but this time from the left network branch's perspective. By definition $k_y^{\text{left}} = 0$ in this case and so all components of Equation 26 fall away, leaving just Equations 25 and 27.

26 and 27.

$$\Omega_y\text{-Norm} = \left( \frac{2^{n_x} n_y \pi_1^2(t)}{k_y^{\text{left}} r^2 + 2^{n_x}} \right)^{\frac{1}{2}} \tag{25}$$

$$\Gamma_{y\text{-}left}\text{-Norm} = \left( \frac{k_y^{\text{left}} n_y r^2 \pi_1^2(t)}{k_y^{\text{left}} r^2 + 2^{n_x}} + (n_x - n_y)\pi_2^2(t) + (2^{n_x} - n_x)\pi_3^2(t) \right)^{\frac{1}{2}} \tag{26}$$

$$\Gamma_{y\text{-}right}\text{-Norm} = \left( n_x \pi_2^2(t) + (2^{n_x} - n_x)\pi_3^2(t) \right)^{\frac{1}{2}} \tag{27}$$

## G  CMNIST ARCHITECTURE AND HYPER-PARAMETERS

In this section we provide the network architectures for the CMNIST experiments as well as other details of the experimental setup. We scale the non-systematic output labels to help the network learn these labels. For the results of this section a scale of 10 was applied, however, the results are consistent for a wide range of scale values. Increasing or decreasing the scale merely changes the time taken for the same effects to occur. Both the dense and split networks are trained using stochastic gradient descent from random initial weights sampled from an isotropic normal distribution. No regularization, learning rate decay or momentum is used. We aim to keep the setup as simple as possible while reducing the effects of other implicit or explicit regularizers on the results, since we are comparing the systematic generalization of the networks. The simplicity also aids the comparison between the dense and split network architectures which is the goal of the experiment. Table 1 shows the hyper-parameters used to train both networks, which have the same hyper-parameters, and the two architectures are shown in Figures 9 (dense architecture) and 10 (split architecture).

Table 1: Table showing the hyper-parameters used for the CMNIST experiments.

| Hyper-parameter | Value |
|---|---|
| Step Size | $2e^{-3}$ |
| Batch Size | 16 |
| Initialization Variance | 0.01 |

We describe some further observations. Consistent with results in linear networks, when training a dense network there is a portion of the systematic mapping which is learned at the same time as the non-systematic mapping (particularly from around epoch 150). In contrast, when using a split architecture the systematic mapping is learned independently of the non-systematic mapping and faster, while the non-systematic mapping struggles to learn. The non-systematic mapping fails to reach a near-zero error and plateaus around 0.4 regardless of how long training lasts. Thus, without a systematic mapping sharing the same hidden layer and helping learning, the non-systematic mapping is ineffective even at fitting the training data in a reasonable amount of time.

Turning to Figure 5b, we see that for both the dense and split networks the non-systematic mapping does not generalize to test data. This is to be expected as it is unlikely that the same numbers in the training data would ever appear in the test data. Notably, comparing the systematic mappings we see that the converged dense network generalizes far worse than the converged split network. It is interesting to note that initially both networks generalize equally well, indicating the potential for IL to be partially effective for a dense network. However, in agreement with our theoretical findings, IL on the dense network would not come without a cost since the network is not able to fully learn the systematic mapping while still maintaining a low generalization error. Since the final portion of the systematic mapping is learned at the same time as the non-systematic mapping, the earliest epoch during training when IL can fully preserve the systematic labels will already have learned some of the non-systematic mapping. However, by learning the non-systematic mapping

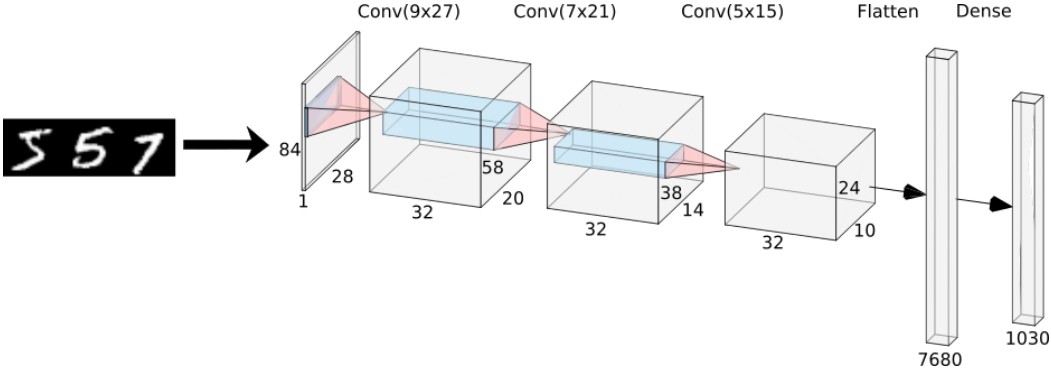

Figure 9: Dense network architecture trained to perform the CMNIST classification task.

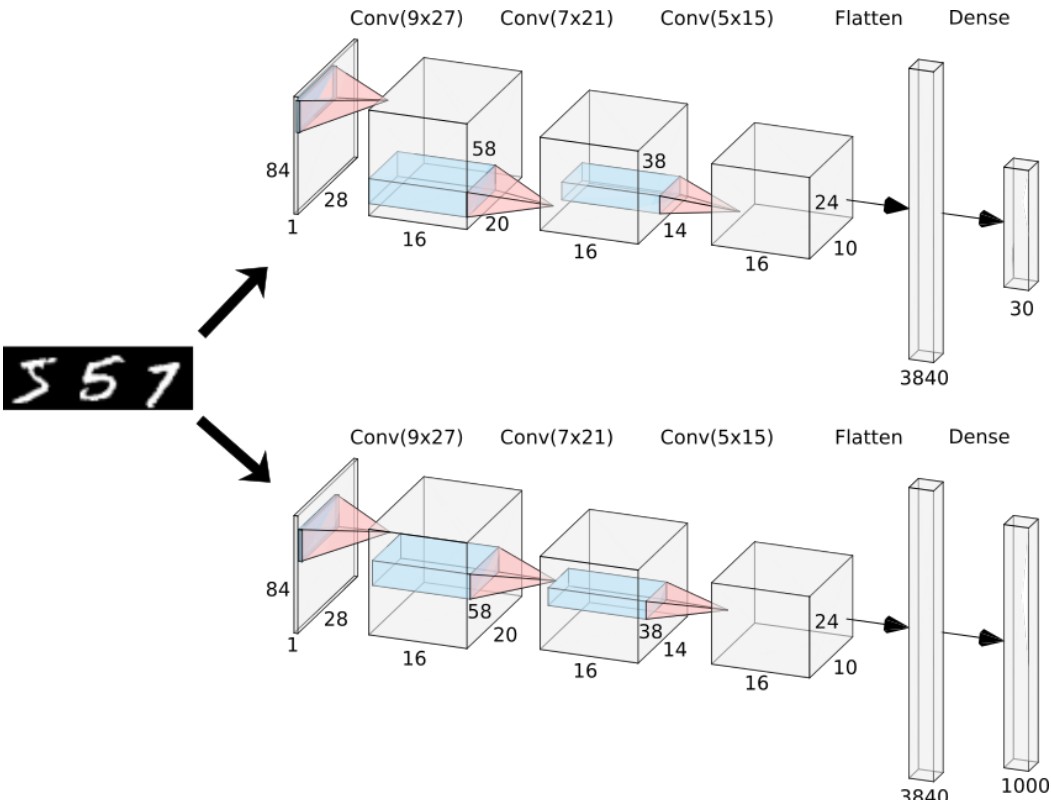

Figure 10: Split network architecture trained to perform the CMNIST classification task.

the network's hidden layer will become worse for generalization, even for the learned systematic mapping. In contrast, the split network architecture sees no generalization gap and maintains a near-zero test error from early on in the training. Thus, it is clear that the benefit of using the split network architecture is that it avoids the conflict in its hidden layers from learning to accommodate both the systematic and non-systematic mappings.

