# OpenReview forum: "The Role of Learning Regime, Architecture and Dataset Structure on Systematic Generalization in Simple Neural Networks"
_ICLR.cc/2022/Conference — ICLR 2022 Submitted_

### Official Review · Reviewer_9QCY · 2021-11-01

**Correctness:** 3
**Technical Novelty And Significance:** 2
**Empirical Novelty And Significance:** 2
**Recommendation:** 5
**Confidence:** 3

**Main Review:**

Strengths
========
The paper investigates the subject of systematic generalization in neural networks, which is important and not well understood yet.
It builds on existing research, and performs reasonable approximations and simplifications in the setting: no non-linearities, synthetic low-dimensional data, gradient descent with a constant learning rate.

Metrics

Weaknesses
==========
Clarity could be improved on different levels:
- The research questions are not clearly stated, and neither are the conclusions in the "discussion" section
- The links between systematicity, the reported metrics, and the general principle of generalization are not obvious. The influence of the non-systematic inputs are quantified, but does such an influence mean that generalization is worse? A network might have learn a way to represent non-systematic inputs in a systematic way in an intermediate layer, I would expect that to be _desirable_. Why would "ignoring the non-systematic inputs" be the right way to generalize?
- It's unusual (and maybe misleading) to refer to 1-hidden-layer neural nets as "deep". There may not be a formal definition, but I'd say 3 hidden layers would be a minimum for "deep" networks.
- It is not clear what questions the experiments on CMNIST are addressing, or what the answers are. Would the iterated learning procedure be applicable in that case?
- I'm not sure what the point of the brand logos vs. brand names are in Figure 1b. Is there a difference between the two? Aren't these labels represented as 1-hot vectors in any case? The caption of Figure 1 mentions that "the name of objects change [...] over generations", but I did not catch any reference to "names" or "spelling" in Section 6.

The paper introduces a whole family of datasets, with factors of variations. However, I don't see them explored, although it would be helpful to see if some results are specific to the variant used. Figure 2 and Figure 4 use different variants, but I'm not sure what's the impact. Similarly, the size of the hidden layer (when there is one) may have an impact on the learning dynamics, and it may be good to address it.

The end of section 4 mentions "the bias the implicit bias arizing from [...] gradient descent". It might be useful to compare against stochastic gradient descent, because it is often argued that SGD has an implicit bias encouraging better-generalizing solutions (compared to full-batch GD).


**Summary Of The Paper:**

The paper introduces a synthetic dataset where examples are composed of both "systematic" inputs (indicating whether one specific feature is present) and "non-systematic" ones (indicating whether a specific combination of all features are present), and where the targets are a subset of systematic inputs, plus all non-systematic ones.
It then studies the training of a few neural network architectures in terms of how the two parts of the inputs are used in predicting each part of the output, theoretically and in practice.

**Summary Of The Review:**

Overall, the reported results are interesting, but the findings are too few and too narrow.
This paper does provide a benchmark and preliminary findings, that could be interesting to build on, but I think this paper does not have enough substance yet to be published at a conference.

---

> ### Author Response · Authors · 2021-11-17
> **Response to Official Review of Paper4134 by Reviewer 9QCY (Part 1 of 2)**
>
> We thank the reviewer for their comments and suggestions. We will address the primary concerns first and then the clarity issues. Firstly the reviewer states: "The paper introduces a whole family of datasets, with factors of variations. However, I don't see them explored...". All equations in this work are written in terms of the five dataset parameters. Thus, all conclusions or proofs using these equations are general to the entire space (or whole family) of datasets. Thus, every point in the space is simultaneously being explored. The reviewer elaborates saying "...although it would be helpful to see if some results are specific to the variant used. Figure 2 and Figure 4 use different variants, but I'm not sure what's the impact". The results are not specific to any variant of the dataset used. We prove in Appendix A that the modes of variation are learned in a particular order regardless of the dataset setting and reference this proof in the main text, demonstrating the generality of our results to the entire space of datasets. Figures 2 and 4 use different dataset settings for visual purposes only. Since we theoretically proved our results hold for the entire space we were free to choose settings for the figures where the conclusion could be easily seen and understood by the reader.
>
> The reviewer also states: "Similarly, the size of the hidden layer (when there is one) may have an impact on the learning dynamics, and it may be good to address it.". The size of the hidden layer does not affect the dynamics as long as it is sufficiently large to still capture all modes of variation. Thus as long as there are at least $2^{n_x}$ (the number of singular values in the input-output covariance) hidden neurons the hidden layer, size will have no effect. If this is not the case then the model cannot learn the desired mapping perfectly and will learn only the top $n_h$ singular values, where $n_h$ is the number of hidden neurons (Plaut, 2018). We agree that this is a potentially interesting case and will add this point to the paper.
>
> Thirdly the reviewer states: "The end of section 4 mentions 'the bias the implicit bias arizing from ... gradient descent'. It might be useful to compare against stochastic gradient descent, because it is often argued that SGD has an implicit bias encouraging better-generalizing solutions (compared to full-batch GD)". The dynamics of learning for the full-batch case will be the same as the stochastic GD case when a sufficiently small learning rate is used (Saxe et al.,2019). This is the case in which our analysis and experiments were conducted as stated in Section 4. We can add additional experiments where we use SGD to empirically verify our results if the theoretical analysis from Saxe et al. (2019) is not sufficient.

---

> > ### Author Response · Authors · 2021-11-17
> > **Response to Official Review of Paper4134 by Reviewer 9QCY (Part 2 of 2)**
> >
> > For the clarity issues the reviewer states:
> > - "The research questions are not clearly stated, and neither are the conclusions in the "discussion" section". We accept this criticism and will endeavour to improve upon this for the final version. Specifically we will add to the second last paragraph of the introduction where we outline the research and our contributions.
> > - "The links between systematicity, the reported metrics, and the general principle of generalization are not obvious. The influence of the non-systematic inputs are quantified, but does such an influence mean that generalization is worse? A network might have learn a way to represent non-systematic inputs in a systematic way in an intermediate layer, I would expect that to be desirable. Why would "ignoring the non-systematic inputs" be the right way to generalize?". To see the problem with relying on non-systematic features we would direct the reviewer to Figure 5 of our own work where we show that by connecting non-systematic features to the systematic output the generalizability of the output drops dramatically (normalized error rises by 0.7 from the green curve, no reliance on non-systematic features, to the blue curve, reliance on non-systematic features). Additionally, a wealth of prior work on systematic generalization has discussed why a reliance on non-systematic features is not desired. As Reviewer 2WAW points out: "Systematic generalisation in neural networks is not a new topic and there has been a lot of work on this subject. E.g., Hadley, Christansen, Chater, Niklasson, Phillips, Gelder, Smolensky...". A further discussion on the relationship between our work and these prior works can be found in the general comment above. In particular the statement "A network might have learn a way to represent non-systematic inputs in a systematic way in an intermediate layer" is not possible for linear or non-linear networks.
> > - "It's unusual (and maybe misleading) to refer to 1-hidden-layer neural nets as "deep". There may not be a formal definition, but I'd say 3 hidden layers would be a minimum for "deep" networks.". The terminology of deep or shallow is chosen based on the network training dynamics. As shown in Saxe et al. (2019) the addition of even one hidden layer changes the network learning dynamics and the addition of hidden layers beyond that does not change the shape of the dynamics. Thus, deep vs shallow are technical terms taken from that previous work referencing whether a network follows the learning dynamics of Equation 3 (deep) or Equation 10 (shallow).
> > - "It is not clear what questions the experiments on CMNIST are addressing, or what the answers are. Would the iterated learning procedure be applicable in that case?". The CMNIST experiment demonstrates that even on this simple dataset a dense neural network fails, which was **predicted** by our theoretical analysis on linear networks. We then present the solution of using modular architectures which was also **predicted** by our theoretical analysis. We would like to emphasize that by our analysis predicting these effects on the convolutional NN with ReLU activation we have demonstrated the applicability of our work to the wider ML community. Thus the CMNIST experiment was addressing the question of whether or not the analysis of the linear networks were applicable to more complex, non-linear networks, on a real-world dataset. As predicted by our analysis, iterated learning would only be applicable (where we are taking applicable to mean fully effective) with the split network architecture. Figure 4 demonstrates this by showing the negative effect including the non-systematic features in the systematic mapping has on the generalization of the network (seen by comparing the blue and green curves). It is also seen in Figure 4(a) by the blue curve only being able to converge once the orange curve reaches a sufficiently low value.
> > - "I'm not sure what the point of the brand logos vs. brand names are in Figure 1b. Is there a difference between the two? Aren't these labels represented as 1-hot vectors in any case? The caption of Figure 1 mentions that "the name of objects change ... over generations", but I did not catch any reference to "names" or "spelling" in Section 6". Figure 1 relates the setup of our dataset to the linguistics literature (Kirby, 2001; Kirby et al., 2008) where iterated learning was first proposed and studied. The car brand logos and names are both represented by identity matrices, however, have different frequencies (logo frequency is determined by $k_x$ and name frequency by $k_y$). In section 6 we discuss the analytical results of our work where we use the space of dataset we have created. Thus no mention is made to the linguistics analogy as it was used earlier in the paper to support our setup and provide intuition.

---

> > > ### Author Response · Authors · 2021-11-17
> > > **Response to Official Review of Paper4134 by Reviewer 9QCY (References)**
> > >
> > > Saxe, Andrew M., James L. McClelland, and Surya Ganguli. "A mathematical theory of semantic development in deep neural networks." Proceedings of the National Academy of Sciences 116.23 (2019): 11537-11546.
> > >
> > > Plaut, Elad. "From principal subspaces to principal components with linear autoencoders." arXiv preprint arXiv:1804.10253 (2018)
> > >
> > > Kirby, Simon. "Spontaneous evolution of linguistic structure-an iterated learning model of the emergence of regularity and irregularity." IEEE Transactions on Evolutionary Computation 5.2 (2001): 102-110.
> > >
> > > Kirby, Simon, Hannah Cornish, and Kenny Smith. "Cumulative cultural evolution in the laboratory: An experimental approach to the origins of structure in human language." Proceedings of the National Academy of Sciences 105.31 (2008): 10681-10686.

---

> ### Comment · Reviewer_9QCY · 2021-11-23
> **Thanks for the response**
>
> I appreciate the detailed response, and I better understand some points in the submission.
> I will try to address the main ones.
>
> * Regarding the parameters of the family of datasets I understand that the proofs provided are generic and the formulas are given in terms of these parameters. However, there are some cases that seems to rely on numerical simulation instead: Section 5 mentions "the input partitioned network", and the Iterated Learning experiment on Figure 4(b) does not seem to have an associated theoretical result. Could it be that Figure 4(b) might look qualitatively different with another dataset in that space?
>
> * Regarding the size of the hidden layer, thanks for explaining it does not matter above a certain threshold, and I appreciate the author's willingness to mention that things may be different for smaller values.
>
> * Regarding SGD, the setting that has been argued to improve generalization is indeed when the learning rate is larger than the infinitesimal limit (where it is equivalent to batch GD), even as large as possible without diverging.
>
> >  "In particular the statement "A network might have learnt a way to represent non-systematic inputs in a systematic way in an intermediate layer" is not possible for linear or non-linear networks.
>
> I'm puzzled by that claim. Maybe the learning dynamics of gradient descent in a linear network do not reach that solution, but I don't see why deep non-linear networks, potentially with other training algorithms, could not learn such a mapping. Maybe that's covered by earlier work on systematicity, though. If that is the case, please let the AC ignore that point.
>
> * Regarding the experiment in Section 7, I think it is beneficial to reframe it, like the authors did in the replies, whithin a research question and with a prediction that the "dense" (non-split, though it's confusing to refer to a CNN as "dense") network will also fail to generalize.
>
>   That being said, the reported experiment on CMNIST has too many limitations to be more than an anecdote or illustration. In particular:
>   - The "normalized error" setting is peculiar, and the loss function itself was not specified (I suppose it's the mean squared error like in the linear case, which would be atypical for a classification problem)
>   - It only addresses one specific architecture, with one specific capacity
>   - The fact that, In Figure 5(a), the train error of the blue and orange curves decrease at the same time during epochs 150-200 is not evidence that "the blue curve cannot converge until the orange curve has become sufficiently low", especially when the test error of the blue curve is going the other way.
>
> Given the other reviews and their replies, I will maintain my recommendation.

---

> > ### Author Response · Authors · 2021-11-23
> > **Comment on Rebuttal Response**
> >
> > We thank the reviewer for engaging. Regarding the first comment on Figure 4(b), the empirical results of the “input partitioned network” did not change any of the conclusions in our work, however, it seemed prudent to mention that this form of partitioning could not be analyzed in the same way as the output partitioned network. Thus, we mention it briefly in Section 5. However, this architecture is not as interesting to the iterated learning framework since it does not achieve output systematicity, which is more applicable to IL since IL involves relabeling the dataset. On the point of whether there could be qualitative differences in Figure 4(b), yes qualitatively there may be differences. However, as with all other empirical results in this work, we are demonstrating one instantiation of a result which is proved for our entire space of datasets. As a result, whatever the qualitative differences may be, the conclusions will be the same. The dense and shallow network will be unable to achieve full systematicity even with IL, the split network can achieve full systematicity when paired with IL and lastly the split network will be as systematic or more so than if optimal early-stopping were used.
> >
> > We thank the reviewer for their advice on Section 7, and hope the new version remedies the previous concerns raised. On the limitations of CMNIST. Firstly, the use of the normalized mean-squared error. Indeed this metric is not common, however, we only use this metric for comparison within our own work (the network is trained with MSE but the normalization is only used for the plots. Training the network with MSE is appropriate since this is the loss function used for our analysis.). We believe this also answers your third point: “The fact that, In Figure 5(a), the train error of the blue and orange curves decrease at the same time during epochs 150-200 is not evidence that...”, since this effect of the systematic error not being able to converge until the non-systematic error is sufficiently low is not observed in the split network. In this case the **only** difference was the splitting of the hidden layer, and this was sufficient to remove the observed convergence effect. Thus, since the normalized MSE is only being used in a comparative, controlled fashion, we believe it is admissible and fair to draw conclusions with it. The fact that the systematic test error increases while the systematic training error has plateaued supports our conclusions, as this points to a separate effect pushing the test error upwards. It appears to be due to the decrease in the non-systematic training error which occurs at the same time, however, we cannot claim this yet. We can make the conclusion of the shared-hidden layer being responsible for this fact, however, when we compare to the split network where these effects are no longer present. As for the limitation of one architecture being used, we would argue that this is a trait of empirical work, and in light of the fact that this section aims to demonstrate the applicability of our theoretical work to a more complex setting we do not think this to be a large weakness.
> >
> > On the point of a network learning a systematic hidden representation from a purely non-systematic input, there is evidence from the disentanglement literature (which we may draw from in discussing latent representations) that this is not possible without supervision (Locatello, 2019). Naturally, the hidden layer representation depends on the down-stream task, however, without hand-crafted features or explicit supervision towards systematicity, it appears that a network will never learn to inject systematicity of its own accord.
> >
> > Locatello, Francesco, et al. "Challenging common assumptions in the unsupervised learning of disentangled representations." international conference on machine learning. PMLR, 2019.

---

### Official Review · Reviewer_2WAW · 2021-11-01

**Correctness:** 4
**Technical Novelty And Significance:** 2
**Empirical Novelty And Significance:** 2
**Recommendation:** 3
**Confidence:** 4

**Main Review:**

The publication has a good motivation but lacks in presentation and results. Systematic generalisation in neural networks is not a new topic and there has been a lot of work on this subject. E.g., Hadley, Christansen, Chater, Niklasson, Phillips, Gelder, Smolensky have in my opinion done very similar analysis in the early 90s. Thus, I find the results up to page 6 not particularly novel or interesting. The authors introduction and related work section consists mostly of very recent work so a more systematic reading of the work following Fodor and Pylyshyn's critique may be adequate.

That said, I found the iterated learning section interesting, albeit rather brief and I'm not sure I fully understood the setup. At which point is training halted and why? Are the logits used to train the next generation or the feature classes? The iterated learning method describes how the description of an observation refines into a more structured language as it is passed through several participants/generations. But in the presented setup, the output neurons of the model have specific meaning since they are tied to specific features. It is not clear to me how this conundrum is resolved.

What are the non-systematic network predictions in the cmnist experiments? The split network is more biased towards separate functions, it's not surprising to me that this built-in bias results in a more systematic function.


**Summary Of The Paper:**

This work aims to study the influences that lead to the systematic generalisation in linear models. They theoretically and empirically explain why a linear model doesn't converge to a systematic solution. They also study iterated learning and find that that with sufficient output modularity the models do converge to systematic solutions.


**Summary Of The Review:**

I find that the paper tackles some interesting questions but does not provide enough value to be accepted. The presentation is lacking. The figures could be improved and the language could be more precise in places. I also find that the work fails to connect with early work on (systematic) generalisation in connectionist models.

---

> ### Author Response · Authors · 2021-11-17
> **Response to Official Review of Paper4134 by Reviewer 2WAW**
>
> We thank the reviewer for their comments and suggestions. We will first address the one critique which states: "Systematic generalisation in neural networks is not a new topic and there has been a lot of work on this subject. E.g., Hadley, Christansen, Chater, Niklasson, Phillips, Gelder, Smolensky have in my opinion done very similar analysis in the early 90s. Thus, I find the results up to page 6 not particularly novel or interesting. The authors introduction and related work section consists mostly of very recent work so a more systematic reading of the work following Fodor and Pylyshyn's critique may be adequate.". We thank the reviewer for the helpful citations and will add them to our work. However, the majority of these prior works are not mathematical analyses of learning in neural networks (Hadley, 1993, 1994; Pollack, 1990; Niklasson, 1992), and the works which are leave the role of architecture and dataset in learning systematic mappings as an open question (Smolensky, 1990; Phillips et al., 1993). Thus we believe the topic is still open to further investigation and elaborate more on this point in the general comment above. Indeed, Reviewer 9QCY states: "The paper investigates the subject of systematic generalization in neural networks, which is important and not well understood yet.". We believe this motivates our work since we do treat systematic generalization theoretically, in the tractable domain of linear networks. We do however also show that our theoretical findings generalize to non-linear convolutional NNs. Thus we find the statement "Thus, I find the results up to page 6 not particularly novel or interesting" confusing since the theoretical treatment presented in the pages prior to page 6 is novel. If the reviewer could point us to where the same theoretical points as ours are made elsewhere we would greatly appreciate it.
>
> We then move on to the points of clarification. The reviewer asks, in reference to IL, "At which point is training halted and why?". Training is halted at a set point prior to the convergence of the network. For the theoretical analysis we leave this point unspecified as our equations are general to accommodate any stopping point. Practically this point is treated as a hyper-parameter similar to early stopping (which is why early stopping is a benchmark in Figure 4(b)). Secondly the reviewer asks: " Are the logits used to train the next generation or the feature classes?". The logits are used and we will clarify this point in the revised edition. Thirdly the reviewer asks: " The iterated learning method describes how the description of an observation refines into a more structured language as it is passed through several participants/generations. But in the presented setup, the output neurons of the model have specific meaning since they are tied to specific features. It is not clear to me how this conundrum is resolved.". The reviewer is correct in that the output neurons have particular meaning. Nonetheless these neurons are refined (all data points have a value of 0 for this feature, or the value is at least decreased towards 0) using IL. This could be seen as non-compositional aspects of language being removed.
>
> In reference to the CMNIST experiment the reviewer asks: "What are the non-systematic network predictions in the cmnist experiments?". In section 7 we state: "the on-systematic output encodes the number as a whole with a 1000-dimensional one-hot vector". The error with the word "on-systematic" which should be "non-systematic" is the likely cause of this clarity issue. We will rectify this error in the paper and thank the reviewer for drawing our attention to it.
>
> Finally the reviewer states: "The split network is more biased towards separate functions, it's not surprising to me that this built-in bias results in a more systematic function.". While the conclusions may be intuitive, in hindsight or otherwise, they are not trivial. The point of theoretical research is to formalize, but also **simplify**, difficult problems for analysis. We do just this. The reviewer also understates our contribution. We do not just show that split networks are more biased towards separate functions (compared to dense networks). We show that split networks are the **only** network biased towards separate functions. By extension we show that dense networks cannot be fully-systematic. Thus it is not a case of one model being more systematic, but which network even has the theoretical capability. As Reviewer ZgRW states in reference to our work: "Understanding and guaranteeing systematicity is an important problem in the design of neural network models. Moreover, a negative result like this is valuable in demonstrating that systematicity is hard in a technical sense".

---

> > ### Author Response · Authors · 2021-11-17
> > **Response to Official Review of Paper4134 by Reviewer 2WAW (References)**
> >
> > Hadley, Robert F. "Systematicity in connectionist language learning." Mind & Language 9.3 (1994): 247-272.
> >
> > Hadley, Robert F. "Connectionism, explicit rules, and symbolic manipulation." Minds and machines 3.2 (1993): 183-200.
> >
> > Pollack, Jordan B. "Recursive distributed representations." Artificial Intelligence 46.1-2 (1990): 77-105.
> >
> > Niklasson, Lars, and Noel Sharkey. Systematicity and generalisation in connectionist compositional representations. Högskolan Skövde/University of Skövde, 1992.
> >
> > Smolensky, Paul. "Tensor product variable binding and the representation of symbolic structures in connectionist systems." Artificial intelligence 46.1-2 (1990): 159-216.
> >
> > Phillips, Steven, and Janet Wiles. "Exponential generalizations from a polynomial number of examples in a combinatorial domain." Proceedings of 1993 International Conference on Neural Networks (IJCNN-93-Nagoya, Japan). Vol. 1. IEEE, 1993.

---

> > ### Comment · Reviewer_2WAW · 2021-11-19
> > **Rebuttal Response**
> >
> > Thank you for your lengthy rebuttal. You might also be interested in section 4 of Steven Phillips' thesis called Connectionism and the Problem of Systematicity from 1995. In that section he compares the principle components of the hidden representation of an MLP trained on a similar compositional task. He also demonstrates (the trivial case) of when a systematic representation does emerge in a dense representation. I think you'll be likely interest in that work given the similarity with the more indirect performance based approach taken in this manuscript. This is one example of the sort of "systematicity in a dense representation" analysis that has been done before and was very popular in the 90's given the harsh critique from philosophers, cognitive scientists, and symbolic AI researchers. Your analysis using linear models might not be exactly the approach of previous work but it is very similar in insight and limitation.
> >
> > In conclusion, I'm sorry to write that I remain unconvinced. In my opinion the current version of the paper lacks novelty and insight for a top venue such as ICLR. I've read your response carefully and I'll take your points into the discussion with the other reviewers.

---

> > > ### Author Response · Authors · 2021-11-22
> > > **Comment on Rebuttal Response**
> > >
> > > We thank the reviewer for engaging. Respectfully, we again point out that this very interesting work (which we will cite) relies on simulation and does not include analytical solutions for the learning dynamics. Many works have applied analyses like PCA to simulation results to attempt to understand the emergence of systematicity. Our work is fundamentally novel in providing mathematical demonstrations of the full learning trajectory for these tasks. With regard to novelty, we urge the reviewer to identify any work which achieves this before ours.
> > >
> > > This makes our work distinctly different compared to prior work. For instance, we can make statements about **any** dataset in our class, which is not possible via simulation. Further, our work points out the specific dataset statistics that lead to systematicity failures, and the specific ways that architectural choices impact both learning dynamics and systematicity. We believe these relationships between task, architecture and learning dynamics have not been formalized before. We do not claim that our work is without limitations, of course, but believe it opens up a dramatically different analytical approach compared to prior work, with many possibilities for future work to build on. We further emphasize that none of the prior work mentioned in the review deals with iterated learning, and we can only make theoretical statements about iterated learning via the novel complete solution trajectories for these tasks that we derive here.

---

### Official Review · Reviewer_u1Hs · 2021-11-08

**Correctness:** 3
**Technical Novelty And Significance:** 2
**Empirical Novelty And Significance:** 2
**Recommendation:** 5
**Confidence:** 3

**Main Review:**

The paper aims at investigating the impact of implicit biases on the learned network mappings both with the means of theoretical and empirical analysis. The impact of prior assumptions is a fundamental problem in learning compositionality. I understand that in order to have full control over the studies system the authors decided on the use of a synthetic dataset, which allows them to derive exact training dynamics.

However, one of my concerns lies exactly with the construction of this dataset and its relevance to the studied problem. The dataset is built from artificial systematic features, which are just all binary combinations over a specified number of bits. The non-systematic features are represented by identity matrixes, where an entry on the diagonal corresponds to a combination of “features”. The target representation follows the exact same definition, with the difference only in the number of sampled “features”. The task, if I understand correctly, is to compute a mapping from the input to the output features. However, without any additional bias or task constraints, it is unclear why a systematic mapping between those features should be favored or even desired. In this light, the observation that such a mapping is not learned, i.e. that non-systematic inputs contribute to the systematic outputs, seems just like an observation of the non-uniqueness of the problem (for example, one could learn something close to an identity mapping or use the non-systematic features as a basis for the outputs. Without any additional biases both those solutions seem to be good fits, so why unsystematic mappings should not be desired in such a case?). Similarly, the observations about the impact of the network architecture (or rather the lack of such) look quite straightforward, and again, without any prior assumptions, it is unclear why the network should even partition the computation.

Finally, beyond the linear study, litter commentary is given on how to address the nonlinear case, apart from a simple CMNIST experiment. I understand that the focus of the paper was on the linear case, however, it questions how impactful the findings are (especially in the light of the above critique) in real-world scenarios.

Technical questions: What is the use of k_x>1? It seems to me that for any k_i, the same identity matrix is used, so the information is not even redundant, but simply the same.

Minor Issues:
Lack of citations in line 8 of the second paragraph in Introductions.
Figure 3: The caption seems not to be on pair with the image (it mentions rows while there are none in the image).

**Summary Of The Paper:**

The paper studies the relation of learning dynamics and architecture structure to the acquisition of systematic knowledge in shallow and deep linear neural networks. To this end the authors study the network behavior on a space of parametric datasets composed of systematic and non-systematic features and assess the impact of training regime, architecture, and iterated learning on the learned by the network functions. It is observed that a fully systematic mapping, thought theoretically possible to learn in such a regime, is not achieved by the shallow nor deep networks, and is not encouraged by a modular architecture (except for the most trivial, fully separated case).  Authors also observe that, in the studied framework, the iterated learning can lead to systematic functions when combined with the output modularity constraint.

**Summary Of The Review:**

As the problem of the role of implicit biases in the systematic generalization seems very interesting, some incorporated constructions and problem designs seem artificial, and the conclusions lack significance, especially in the light of their limitation to the linear case study.

---

> ### Author Response · Authors · 2021-11-17
> **Response to Official Review of Paper4134 by Reviewer u1Hs**
>
> We would like to thank the reviewer for their comments and suggestions. A first primary critique appears to be that it is unclear why relying on non-systematic features is not desired. To see the problem with relying on non-systematic features we would direct the reviewer to Figure 5 of our own work where we show that by connecting non-systematic features to the systematic output the generalizability of the output drops dramatically (normalized error rises by 0.7 from the green curve, no reliance on non-systematic features, to the blue curve, reliance on non-systematic features). Additionally, a wealth of prior work on systematic generalization has discussed why a reliance on non-systematic features is not desired. As Reviewer 2WAW points out: "Systematic generalisation in neural networks is not a new topic and there has been a lot of work on this subject. E.g., Hadley, Christansen, Chater, Niklasson, Phillips, Gelder, Smolensky...". We are, to the best of our knowledge, one of the first to treat the learning of systematicity theoretically using mathematical analyses of gradient descent dynamics in neural networks. A further discussion on the relationship between our work and these prior works can be found in the general comment above.
>
> A second primary critique appears to be the intuitive nature of our conclusion. In particular from statements such as "Similarly, the observations about the impact of the network architecture (or rather the lack of such) look quite straightforward, and again, without any prior assumptions, it is unclear why the network should even partition the computation" and "...seems just like an observation of the non-uniqueness of the problem". While the conclusions may be intuitive, in hindsight or otherwise, they are not trivial. The point of theoretical research is to formalize, but also **simplify**, difficult problems for analysis. We do just this. In addition, if our conclusions are as intuitive as suggested, then it calls into question why there is even debate around the ability of neural networks to systematically generalize, which is still being discussed as of Dankers et al. (2021). As Reviewer ZgRW states in reference to our work: "Understanding and guaranteeing systematicity is an important problem in the design of neural network models. Moreover, a negative result like this is valuable in demonstrating that systematicity is hard in a technical sense".
>
> A third critique is on the lack of results for the non-linear case aside from the "simple" CMNIST experiment. We would emphasize that the simplicity of the CMNIST result is a strength, since we demonstrate that **even** on this "simple" dataset a dense neural network fails, which was **predicted** by our theoretical analysis on linear networks. We then present the solution of using modular architectures which was also **predicted** by our theoretical analysis. We would like to emphasize that by our analysis predicting these effects on the convolutional NN with ReLU activation we have demonstrated the applicability of our work to the wider ML community. Analyzing non-linear network dynamics falls outside the scope of this work, which analytically treats systematic generalization, iterated learning and demonstrates that these predictions carry empirically to a CNN with non-linear activation on a real-world dataset.
>
> A fourth critique states: "Technical questions: What is the use of k\_x>1? It seems to me that for any k\_i, the same identity matrix is used, so the information is not even redundant, but simply the same." The effect of $k_x$ can be seen in the equations 4 to 9 and 11 to 12. Clearly this variable has an effect on the learning dynamics for values greater than 1, as we demonstrate. If this effect of $k_x$ alone is unintuitive then we would suggest that this is a contribution of our work, as our equations show that repetition of non-systematic features can drive the network to a non-systematic mapping (which reduces generalizability). We also motivate the use of repetition by citing prior work which states that "frequency...can promote non-compositional language being used by humans (Rogers et al.,2004)". Lastly, a primary aim of Figure 1 is to provide some intuition for the kinds of features in our dataset. To elaborate here, non-systematic features are atomic properties of an object which can be used to individuate the object. For example, a car might have a particular logo, hood ornament or even a particular scratch. $k_x$, thus represents the frequency or number of these non-systematic features that are present on a given object and can be used to individuate the object.
>
> We thank the reviewer for pointing out the Minor issues and will correct those for the revised version.

---

> > ### Author Response · Authors · 2021-11-17
> > **Response to Official Review of Paper4134 by Reviewer u1Hs (References)**
> >
> > Dankers, Verna, Elia Bruni, and Dieuwke Hupkes. "The paradox of the compositionality of natural language: a neural machine translation case study." arXiv preprint arXiv:2108.05885 (2021).
> >
> > Timothy T Rogers, James L McClelland, et al.Semantic cognition: A parallel distributed processing approach. MIT press, 2004.

---

> > > ### Comment · Reviewer_u1Hs · 2021-11-23
> > > **Rebuttal Response**
> > >
> > > Thank you for your detailed response. The critique was not directed towards the usefulness of the non-systematic features in general, but rather in the specific dataset scenario presented in Figure 1. I.e. providing $\Gamma_x$ provides at the same time the basis, so given only $\Gamma_x$ one can always recover $\Gamma_y$ and $\Omega_y$, regardless of $\Omega_x$. So in this particular problem, both mappings, systematic and non-systematic will give the exact same results. Thus, in my understanding, they would both, in this situation, generalize. It is, however, interesting to see the way the training dynamics depend on the frequency (i.e. repetition of the information in form of $k_x$, thank you for clarifying that).  I agree that the experiment in Figure 5 is more interesting, as the inputs are from a different space, so this poses an additional “problem” (another bias) of which type of network would be better suited for leveraging such representation. Due to this "another bias" and the described above ambiguity of the mappings from Fig 1. the link between the theoretical analysis and the experiments in Fig 5. (and to any other non-artificial dataset scenario) is very unclear for me.
> > >
> > > Minor note: In this scenario, it would be also interesting to see (empirically) whether the learning curves from Figure 5 change when going deeper with the network.

---

> > > > ### Author Response · Authors · 2021-11-24
> > > > **Comment on Rebuttal Response**
> > > >
> > > > We thank the reviewer for engaging. The dataset in Figure 1 is used as an example to aid in the understanding of our setup and how it might relate to the linguistics literature (mapping from a logical to phonetic form). However, we would like to reiterate that from the equations derived in this work we show that all conclusions are applicable to the entire space of datasets, due to these equations being written in terms of the five parameters which specify the dataset. Thus, we are free to choose example datasets without loss of generality. As for the generalizability of the trained model. The redundancy in the input information as well as the training data being exhaustive of all feature combinations (and by extension all unique identifiers are also seen during training) are a property of all datasets in our space as long as  $n_x, k_x, r > 0$. This is so that we may identify whether the model has a preference towards a systematic, non-systematic or mixed mapping. Thus, by design, the input-output mapping can be learned perfectly while having varying degrees of systematicity. It is the preference exhibited by the model towards systematicity which we investigate. However, a reliance on a non-systematic mapping (the unique identifiers) naturally does not generalize well compared to a more systematic approach as a model is unable to reason about the object based on its constituent pieces or features. This effect is seen clearly in our CMNIST experiments as the test error performance decreases when non-systematic mappings are used, which agrees with the prior literature on this point.
> > > >
> > > > With regard to the link between the CMNIST experiment and our theory, the primary link is that our theory makes a number of predictions about the effect of sharing a hidden layer representation between systematic and non-systematic output labels. We elaborated on this in the general comment above and have updated the paper to improve the explanation of these points. However, in summary, our theory predicts that a model with such a shared representation should see an increase in test error and a slower convergence rate, compared to a split architecture as we propose, even for the systematic mapping. From Figure 5 we see that these predictions are indeed true. Thus, we believe the CMNIST experiment serves the intended purpose of displaying the applicability of our theoretical analysis to a non-linear and real-world domain.
> > > >
> > > > We thank the reviewer for the suggestion on using a deeper architecture for CMNIST. It is exactly these kinds of questions and extensions which we believe the CMNIST dataset will lead to and hopefully be used to shed light on in future work. Unfortunately, a fuller investigation into the many different architectural choices available falls outside of the scope of this work. Thus, we picked a sufficiently deep architecture to be realistic (at least for a dataset like MNIST) for this experiment.

---

### Official Review · Reviewer_ZgRW · 2021-11-09

**Correctness:** 3
**Technical Novelty And Significance:** 3
**Empirical Novelty And Significance:** 2
**Recommendation:** 5
**Confidence:** 4

**Main Review:**

#### Strengths

1. Understanding and guaranteeing systematicity is an important problem in the design of neural network models. Moreover, a negative result like this is valuable in demonstrating that systematicity is hard in a technical sense.

1. The submission formalizes a simple data setting in which systematicity is controllable. Controlling systematicity in the dataset in this way is a nice approach that I haven't seen taken before.

#### Weaknesses

1. Simplifying assumptions limit the applicability of the theoretical insights to standard machine learning pipelines: binary features, deep linear nets. In addition, the empirical study in Section 7 is not tied closely to the theoretical investigations, so its significance is not clear.

1. The formalization is adapted from the proof technique of Saxe et al. (2019), so the technical novelty there is smaller, although the theoretical goal is distinct.

1. The paper could use many improvements in clarity in critical sections and several attributions to prior work (see minor comments below). In addition, many critical explanations are relegated to the Appendix.

#### Minor comments

1. "...however, a number of studies have identified situations where depth alone is insufficient for structured generalization ()" missing reference?
1. "If systematic components are easier to learn than non-systematic ones, this process [of iterated learning] can successively refine a language toward a systematic structure..." I'm not sure that I agree with this claim in isolation, or at least "learnability" needs to be better defined.
1. "the closely related concept [to systematicity] of compositional structure" Can you give precise definitions of systematicity and compositional structure?
1. "Thus, without regularizers, compositional mappings do not emerge with NMNs." It should also be noted that NMNs require auxiliary data (i.e, the string, parse, etc. associated with a scene).
1. "Iterated learning (IL) approaches..." Iterated learning should be intuitively defined before this point. I also found the rest of this paragraph hard to understand at this point in the paper.
1. I think the clarity of the formalization paragraph could be increased. It was hard to understand what is part of the underlying data-generating process vs. what is given to a model as observation, and also what role is played by the "non-systematic input feature matrix" vs. the "non-systematic output feature matrix". It is not clear what the "frequency and intensity" of non-systematic features means, and similarly for "the amount" of systematic structure—I can believe that these notions should have an effect on the learning dynamics, but these notions are not precisely defined.
1. "This space of datasets is consistent with numerical notions of compositionality in previous works (Andreas, 2018), which define compositionality as a homomorphism between the observation space and the naming space." This should be better explained to argue more clearly for the specific decisions made in the present formalization.
1. "The generalization abilities of deep networks depend on a complex interplay of learning dynamics, architecture, initialization, and dataset statistics." Citations should be added as there is much work on this topic.
1. "...which prior work has shown to impart a low-rank inductive bias on the linear mapping." Specific citation missing.
1. Are Eqs. (1) to (12) from prior work, or newly proven in this paper? Please clarify what is re-used from e.g., Saxe and what are new developments.
1. Before Eq. (1), all assumptions on the specific training setup (architecture, initialization) should be (even briefly) mentioned in the text. I don't think leaving these to the appendix is sufficient, since they are necessary to scope the impact of the result.
1. It is not clearly explained in the text of Section 4.1 what is the significance of the Frobenius norm and singular value trajectories in measuring usage of systematic vs. non-systematic features. Similarly, it is not quite clear at this point what are systematic inputs vs. outputs, and systematic features have not yet been very precisely defined. These clarity issues make it quite hard to understand the implications of the plots in Figure 2.
1. In Section 5, it is not explained how the "output-" and "input-partitioned networks" are constructed. I found interpreting these results extremely difficult because the setup is under-explained.
1. Section 6 is nice in exploring iterated learning as a regularizer, but I also found this section very difficult to understand because it builds on technical structure from prior sections which is insufficiently explained.
1. It is mentioned in Sections 5 & 7 that "[optimal] early stopping" is used; is this using an appropriate validation (not test) split? If not, this optimal early stopping is not achievable in practice, and so there is not evidence that non-systematic features could be eliminated.

**Summary Of The Paper:**

The submission studies what properties of a training setup impact systematicity of a neural network. The theoretical setting uses the deep linear neural networks setup of Saxe et al. (2014; 2019) and derives the effect of the degree of systematicity of a dataset on a model's systematic behavior. The theoretical results demonstrate that systematicity is difficult to guarantee, but that structural modularity and iterated learning are helpful to improve systematicity.

**Summary Of The Review:**

My main concern is the applicability of the insights to more standard settings (W1). The submission also needs major improvements in clarity (W3).

---

> ### Author Response · Authors · 2021-11-17
> **Response to Official Review of Paper4134 by Reviewer ZgRW (Part 1 of 3)**
>
> We thank the reviewer for their comments, and for already taking the time to expand on their review since it was given. We will address the weaknesses and minor comments in order.
>
> ### Weaknesses
> 1\. "Simplifying assumptions limit the applicability of the theoretical insights to standard machine learning pipelines: binary features, deep linear nets.". We see these simple restrictions as a step towards the more complex analysis. Presently linear neural networks are one of the most powerful analytical tools at our disposal, and we chose to use this model as a result. We would argue that the use of linear activations reduces applicability less (for our particular study) than with other analytical tools that rely on infinite width limits. As has been shown by Saxe et al. (2014,2019) linear networks demonstrate a number of aspects of the effect of depth that are also present in nonlinear networks, and feature learning specifically. The same cannot be said for the NTK where feature learning does not occur. Thus, since this work aimed to analyze feature learning effects between shallow and deep networks, we chose the most advanced analytical tool which was appropriate for our aims. We do agree that a more general analytical tool is needed, and see the analysis of linear networks as a step towards this goal, motivating this work. Our use of the simplifying assumptions is supported by Reviewer 9QCY who states in reference to our work: "It builds on existing research, and performs reasonable approximations and simplifications in the setting: no non-linearities, synthetic low-dimensional data, gradient descent with a constant learning rate."
>
> "In addition, the empirical study in Section 7 is not tied closely to the theoretical investigations, so its significance is not clear.". Thank you for highlighting the lack of clarity in this section. We will update the paper and endeavour to fix this issue. To clarify here, CMNIST aimed to show that the same effects from our linear network setup could be seen in a more general setting. We were aiming to demonstrate the applicability of our analysis. In particular, non-binary features were used and ReLU activation was now incorporated with convolutional layers for features extraction. The CMNIST experiment, specifically Figure 5, shows three key results. Firstly that in dense networks the error of the systematic mapping is tied to the error of the non-systematic mapping. This is seen in Figure 5(a) where the blue curve cannot converge until the orange curve has become sufficiently low. Secondly, comparing Figure 5(a) and 5(b) we demonstrate the lack of generalization which occurs when using non-systematic features. This is seen as the orange and red curves achieving a lower training error while the test error increases. Lastly, again by comparing Figure 5(a) and 5(b), we see that the systematic mapping of the split architecture is the only mapping which generalizes well (near 0  training and test error). Thus, we see that even in this more complex setting, the sharing of the hidden layer between systematic and non-systematic features results in a poor hidden representation for generalizability being learned, which was **predicted** by our linear network analysis. We elaborate more on the CMNIST setting in the general comment above.
>
> 2\. "The formalization is adapted from the proof technique of Saxe et al. (2019), so the technical novelty there is smaller, although the theoretical goal is distinct". We do rely heavily on Saxe et al. (2019) for the network dynamics analysis. In particular the only dynamics which are novel to this work is the shallow network **with** input correlations which is not found in Saxe et al (2019). However, as the reviewer points out in the "Strengths" portion of the review: "The submission formalizes a simple data setting in which systematicity is controllable. Controlling systematicity in the dataset in this way is a nice approach that I haven't seen taken before". We see this approach of formalizing the dataset, and using prior analysis as the interpretation tool, as the theoretical contribution. Specifically, the fact that the linear networks dynamics are then written in terms of the dataset parameters, which allows us to analyse networks across the entire scope of available datasets, is the theoretical contribution. In addition we also analyse modular network architectures which Saxe et al. (2019) does not.

---

> > ### Author Response · Authors · 2021-11-17
> > **Response to Official Review of Paper4134 by Reviewer ZgRW (Part 2 of 3)**
> >
> > 3\. "The paper could use many improvements in clarity in critical sections and several attributions to prior work (see minor comments below). In addition, many critical explanations are relegated to the Appendix.". We would like to again thank the reviewer for these helpful comments, and for taking the time to add and elaborate more since the reviewers were released. We will address the minor comments below, however, regarding the relegation of information to the appendix, we are not certain which sections the reviewer is referring to. Most sections in the appendix build off of the results in the main text (examples are merging input or output partitions, which is achieved by merely summing the norm equations correctly) or provide the full equations which were not directly reference in the text but are still helpful. All necessary equations from the appendix are stated or written in the main text. One proof is provided in Section A, however, providing proofs in the appendix is standard practice for this style of paper. If the reviewer could elaborate on certain sections that should be incorporated into the main text from the appendix, we would be happy to do so.
> >
> > ### Minor comments
> > 1. We will rectify this in the updated version.
> > 2. Learnability is in reference to how quickly a mode from the SVD reaches near its asymptotic value. We will clarify this in the text. However, all modes which are not learned before a generation of IL ends will be removed (or whatever proportion is not learned will be removed) and the SV for that mode will tend to 0. Thus, by refining a mode's SV to 0, IL will remove the mode. Which modes are removed, thus, depends entirely on learning speed (what we would say determines how easy it is learn).
> > 3. We will clarify these terms.
> > 4. While NMNs have mainly been used for VQA tasks, we believe these models are more widely applicable. The auxiliary data which "grounds" the data point are there to make the problem more difficult. The difference between SCAN (Lake, 2018) and gSCAN (Ruis, 2020) is the primary example where NMNs have been used in both cases but SCAN does not include auxiliary data, while gSCAN does.
> > 5. In the introduction (before the point being referenced in the Background) we state "Iterated Learning (IL), a method in which generations of agents train briefly on a `language' produced by their parent". We then expand on this in the Background section being referenced in preparation for our own work in Section 6.
> > 6. We will endeavour to clarify this section. Each term the reviewer mentions has a corresponding dataset parameter, where "frequency" is $k_x$ and $k_y$, "intensity" is $r$, and the "amount of systematic structure" is $n_x$ and $n_y$. These notions are all present in the network dynamics and providing the dynamics in terms of these variables is the primary theoretical contribution. With regard to "It was hard to understand what is part of the underlying data-generating process vs. what is given to a model as observation", the matrices $\Omega_x$ and $\Gamma_x$ form the input matrix $X$ (all capital letters). These matrices are parametrized by the lower case variables $n_x, n_y, k_x, k_y$ and $r$ which determine the data generation process. On the top of the second paragraph of page 4 we state "we define a parametric space of datasets with input and output matrices $X= [\Omega_x  \Gamma_x]$ and $Y= [\Omega_y  \Gamma_y]$ respectively, where $n_x,n_y,k_x,k_y,r \in R+$ are the parameters that define a specific dataset".
> > 7. We will add to the clarity of this citation.
> > 8. The quote being referenced "The generalization abilities of deep networks depend on a complex interplay of learning dynamics, architecture, initialization, and dataset statistics." is introducing our own work. This statement summarizes our own contribution in analyzing this interplay. Thus, any relevant citations are given in the Background, and a citation displaying exactly this statement, to the best of our knowledge, does not exist as we are aiming to fill this gap in the literature in this work. We will include the citations from Reviewer 2WAW, but as mentioned in the general comment above, these prior works support our own but there are some key differences. If the reviewer can point us to any additional citation we would be most appreciative.
> > 9. Six citations are given immediately before the statement being referenced. The entire statement is "They therefore serve as a tractable model of the influence of depth specifically on learning dynamics, which prior work has shown to impart a low-rank inductive bias on the linear mapping" where "They" is referring to linear networks studied in these six citations. We will add more citations to this sentence in our work.

---

> > > ### Author Response · Authors · 2021-11-17
> > > **Response to Official Review of Paper4134 by Reviewer ZgRW (Part 3 of 3)**
> > >
> > > 10. Equations 4 to 10 and equations 11 and 12 are all novel. Equations 1 and 2 are merely just singular value decompositions of the covariance matrix, like those used in PCA, and are not novel. Equation 3 for the deep network dynamics with input correlations is from Saxe et al. (2019). To be clear, Equation 10, the shallow network dynamics with input correlations, is new to this work, however we followed the same derivation strategy as Saxe et al. (2019) to arrive at these dynamics, and the shallow dynamics are more broadly known by association to the linear regression dynamics. We will clearly state which equations are new to this work.
> > > 11. Appendix A restates mainly the approach of Saxe et al (2019) in our context and provides the mathematical notation for what *is* stated in the main text "  In particular, "consider a single hidden layer network computing output $\hat{y}=W^2W^1x$ in response to an input $x$, trained to minimize the mean squared error loss using full batch gradient descent with small learning rate epsilon". Figure 3 in the main text provides the network architecture used, however, we will make the reference to Figure 3 explicit in Section 4.
> > > 12. This point is similar to point 3, we will endeavour to provide clearer definitions for systematicity. Since these derivations are achieved by using the SVD of the covariance matrix, the interpretation is the same as for a method like PCA. The singular value depicts how one semantic dimension of the input space maps onto a semantic dimension of the output space. The Frobenius norm then provides a measure of the full (or a subset of the) mapping from the entire input space onto the full (or a subset of the) output space.
> > > 13. In section 5 we point to Figure 3 to aid in the clarity of the architectures. As mentioned by the reviewer Figure 3 references "rows" which do not match the image. The reference to "top row" is meant to be towards the architecture diagrams, while the "bottom row" is meant to be directed towards the graphical representation of our results. We will fix this error in the caption and believe that this will clear up any confusion on the partitioning of the network structure.
> > > 14. Based on the reviewers previous comments we will improve the clarity of earlier sections and hope this will improve the clarity of Section 6 which is a key contribution of this work.
> > > 15. Since we are able to fully analyse the dynamics, we refer to "optimal early stopping" as the theoretical best point to early stop to aid generalization. As the reviewer correctly points out, this is unlikely to ever be achieved in practice. However, this is a good thing for our work as this means we have given the benchmark metric the theoretically optimal conditions and it still does not achieve the same benefits as IL (showing the benefits of IL is a major contribution of this work).

---

> > > > ### Author Response · Authors · 2021-11-17
> > > > **Response to Official Review of Paper4134 by Reviewer ZgRW (References)**
> > > >
> > > > Saxe, Andrew M., James L. McClelland, and Surya Ganguli. "A mathematical theory of semantic development in deep neural networks." Proceedings of the National Academy of Sciences 116.23 (2019): 11537-11546.
> > > >
> > > > Saxe, Andrew M., James L. McClelland, and Surya Ganguli. "Exact solutions to the nonlinear dynamics of learning in deep linear neural networks." arXiv preprint arXiv:1312.6120 (2013).
> > > >
> > > > Lake, Brenden, and Marco Baroni. "Generalization without systematicity: On the compositional skills of sequence-to-sequence recurrent networks." International conference on machine learning. PMLR, 2018.
> > > >
> > > > Ruis, L., et al. "A Benchmark for Systematic Generalization in Grounded Language Understanding." Advances in Neural Information Processing Systems 33 (NeurIPS 2020) (2020).

---

### Author Response · Authors · 2021-11-17
**General Comments to Reviewers (Part 1 of 3)**

We thank all of the reviewers for their comments and suggestions. In this general comment we address what we see as the main criticisms of this work.

## Lack of theoretical significance and novelty
The dependence of this work on that of Saxe et al. (2019) and the similarity of our initial conclusions to prior work has caused Reviewers ZgRW and 2WAW to question the technical novelty of this work. We would like to emphasize that the training dynamics of the linear networks (which is what we use from Saxe et al. (2019)) is not the technical contribution of this work. Rather our contribution is the manner in which we use these dynamics. By formalizing a space of intuitive and semantically meaningful datasets and then solving for the dynamics in terms of the dataset parameters we may draw conclusions about the effect of data on the network. Thus, our contribution is in formalizing the dataset part of the pipeline and the novel manner in which we use the dynamics to formalize how dataset structure affects the dynamics. As Reviewer ZgRW states: "The submission formalizes a simple data setting in which systematicity is controllable. Controlling systematicity in the dataset in this way is a nice approach that I haven't seen taken before". We will aim to make it clearer that this is our primary contribution in the final version. Finally, for a theoretical work this research covers a wide range of topics. In particular we 1. analyze systematic generalization with linear networks, which involves creating a space of datasets and solving the general SVD for the entire space, 2. probe the learning dynamics of iterated learning, 3. expose the necessity of modular architectures for the regularization offered by IL to be completely effective, and 4. demonstrate that our theoretical conclusions generalize to a non-linear CNN trained on a real-world dataset. Thus, the theoretical significance of this work is also spread over its broad range of topics which also increases the applicability of this work to the wider ML community.

Reviewer 2WAW also cited the work of a variety of authors from the early 90s as reducing the novelty of this work, namely "Hadley, Christansen, Chater, Niklasson, Phillips, Gelder, Smolensky". In light of these prior works we may make a few observation, which contextualize this work. Firstly, the vast majority of the prior works are **not** mathematical analyses of deep networks, and rather rely on thought experiments, demonstrations and argumentation to draw conclusions about systematicity. While it is true that these works, particularly Hadley (1993, 1994), Pollack (1990) and Niklasson (1992) point towards the utility of modularity with connectionist networks (with Hadley (1993) being the only one to speak of modularity explicitly); none of these works mathematically show the necessity of modularity for systematicity in a space of datasets. Thus, our work and conclusions are consistent with the hypotheses of these earlier works, however, we now theoretically prove the necessity of modularity for systematicity to emerge in neural networks. Thus, these prior works motivate and strengthen our own work, however, our conclusions (while consistent with prior works) are stronger. Secondly, as far as we can tell, we are the first to include systematic and non-systematic features in our datasets. This added a new consideration not present in the prior works, which focus only in the potential for networks to learn systematic features, not their preference towards systematicity when non-systematicity is a viable alternative. Thus, due to the added degree of nuance this brings to our setup, we show theoretically that connectionist networks are not even weakly systematic, to use the definitions from Hadley (1993), when presented with alternative mappings. This **is** a new conclusion not present in any prior works. We will aim to clarify our contributions relative to these papers in the revision.

---

> ### Author Response · Authors · 2021-11-17
> **General Comments to Reviewers (Part 2 of 3)**
>
> Of the authors mention Smolensky (1990) and Phillips et al. (1993) present two theoretical works. However, Smolenksy (1990) does not consider learning (rather showing how fixed systematic mappings can exist); and Phillips (1993) (which adopts a PAC framework) does not treat the inductive biases in neural networks trained with gradient descent. Specifically, the question of what architectural bias to impose, as well as the role of the environment (dataset) is left as an open question. This alone is the strongest motivation of our current work, as we have now made progress in answering these two questions which have remained open for nearly 30 years (to the best of our knowledge). This also directs to our final point, these prior works which hypothesize about the need for architectural biases all drew conclusions which we now prove to be correct, however, we also show that the architectural bias alone is not sufficient, and that iterated learning with the correct architectural bias is the **only** method in our setup capable of obtaining systematicity. We are gratified that Reviewer 2WAW found the iterated learning portion of our work interesting, but we note that this work **requires** our other results on the role of architecture and the dataset to present a full argument, since all of these biases together are needed for systematicity. Thus, in this work we have also begun to unify the seemingly disparate work on systematicity from the 90s, which is focused on architectural bias and symbolic manipulation, with the work on systematicity from the 2000s, which is focused on iterated learning. We do so by using advances in the theory of neural networks which has only emerged recently. Thus, we believe these prior works strongly motivate the contributions we make in this paper.
>
> ## Applicability of theoretical assumptions and synthetic dataset
> We think the theory of deep learning is extremely important, and these simple linear models and controllable datasets are steps towards the more complex analysis. Additionally, we are pleased that no reviewers identified any technical flaws in this work. Thus, the primary critique on the applicability of our work is the theoretical assumptions. We are pleased that Reviewer 9QCY found our setting reasonable as they state: "It builds on existing research, and performs reasonable approximations and simplifications in the setting: no non-linearities, synthetic low-dimensional data, gradient descent with a constant learning rate".
>
> Importantly we would like to contrast our theoretical setup with another powerful theoretical tool, the Neural Tangent Kernel (NTK), which requires an infinite width limit for network hidden layers. As a result the NTK works in the "lazy-training" regime and removes the feature learning aspect of training. The infinite width limit and lazy regime are both limitations of this widely studied theoretical tool. However, since a key part of our work is on feature learning, linear neural networks are the most powerful **and** appropriate tool at our disposal. Thus, from a theoretical point of view linear networks are no more limited than other accepted methods, and are the appropriate choice in our case. It appears that there is a clear lack of theory in the literature in the direction of formalizing architecture choices, and the impact of dataset on learning. We aim to advance work in this direction and since we are the first to build off of Saxe et al. believe this work is a promising first step.

---

> > ### Author Response · Authors · 2021-11-17
> > **General Comments to Reviewers (Part 3 of 3)**
> >
> > Finally, on the applicability of our space of datasets. We would argue for the applicability on two fronts. Firstly, in this work we motivate our space of datasets by drawing on the linguistics literature, where each of our dataset parameters has a corresponding concept from linguistics where $k_x$ and $k_y$ represents "frequency" in the observation and linguistics structure, $r$ is "intensity" of both the observation and corresponding word (which has to match for language to be useful), and $n_x$ and $n_y$ correspond to the "systematic structure" of an observation and the corresponding language. Figure 1 aims to make the realistic nature of these variables more evident through an example of the cognitive mapping. Importantly, through us aiming to formalize systematic data in this way, other datasets which use language or compositional structure in the inputs or outputs are complex examples of the same kind of dataset we study. Examples of such dataset are SCAN (Lake, 2018) and gSCAN (Ruis, 2020) which are commonly used to investigate systematic generalization. For visual learning tasks the features learned by a CNNs hidden layers are also compositional. Thus, if it is helpful, the binary and one-hot features of our dataset can be thought of as indicating the presence or absence of a visual feature in a similar manner to the feature maps of a CNN. Secondly, simplified binary variables have been used before for more theoretical works which do not aim to advance the state of the art. Notably Baptistsa and Poloczek (2018) use binary variables to propose a novel algorithm for optimization of combinatorial structure, to name one example.
> >
> > ## Significance of CMNIST
> > We acknowledge that the CMNIST experiment description requires more specificity and we aim to improve on this for the revised version. However, this dataset is an appropriate real-world example of the effects predicted by the space of datasets and the linear networks. Primarily, CMNIST has systematic (labeling each of the three digits separately) and non-systematic (one label showing the full value of the number) labels. The effect **predicted** by our theoretical work is that this redundancy in the labels will interfere with the network's learned hidden representations and decrease the systematicity, and generalizability, of the network. This prediction is shown to be true in Figure 5. Specifically, the three primary points of the CMNIST experiment are still in the main body of the text and are evident in Figure 5. Firstly, in dense networks the error of the systematic mapping is tied to the error of the non-systematic mapping. This is seen in Figure 5(a) where the blue curve cannot converge until the orange curve has become sufficiently low. Secondly, comparing Figure 5(a) and 5(b) we demonstrate the lack of generalization which occurs when using non-systematic features. This is seen as the orange and red curves achieving a lower training error while the test error increases. Lastly, again by comparing Figure 5(a) and 5(b), we see that the systematic mapping of the split architecture is the only mapping which generalizes well (near 0  training and test error). Thus, even in this more complex setting, we see the negative effect including the non-systematic features in the systematic mapping has on the generalization of the network (seen by comparing the blue and green curves). These empirical results support the theoretical **prediction** that sharing of the hidden layer between systematic and non-systematic features results in a poor hidden representation for generalizability being learned. We would like to emphasize that the simplicity of this dataset is a strength in light of the fact that a non-linear CNN is still unable to generalize well on this task. Thus, a contribution of this work is in the creation of this dataset which, simply and clearly, exposes this problem with CNNs and can be used for future work.

---

> > > ### Author Response · Authors · 2021-11-17
> > > **General Comments to Reviewers (References)**
> > >
> > > Saxe, Andrew M., James L. McClelland, and Surya Ganguli. "A mathematical theory of semantic development in deep neural networks." Proceedings of the National Academy of Sciences 116.23 (2019): 11537-11546.
> > >
> > > Hadley, Robert F. "Systematicity in connectionist language learning." Mind & Language 9.3 (1994): 247-272.
> > >
> > > Hadley, Robert F. "Connectionism, explicit rules, and symbolic manipulation." Minds and machines 3.2 (1993): 183-200.
> > >
> > > Pollack, Jordan B. "Recursive distributed representations." Artificial Intelligence 46.1-2 (1990): 77-105.
> > >
> > > Niklasson, Lars, and Noel Sharkey. Systematicity and generalisation in connectionist compositional representations. Högskolan Skövde/University of Skövde, 1992.
> > >
> > > Smolensky, Paul. "Tensor product variable binding and the representation of symbolic structures in connectionist systems." Artificial intelligence 46.1-2 (1990): 159-216.
> > >
> > > Phillips, Steven, and Janet Wiles. "Exponential generalizations from a polynomial number of examples in a combinatorial domain." Proceedings of 1993 International Conference on Neural Networks (IJCNN-93-Nagoya, Japan). Vol. 1. IEEE, 1993.
> > >
> > > Lake, Brenden, and Marco Baroni. "Generalization without systematicity: On the compositional skills of sequence-to-sequence recurrent networks." International conference on machine learning. PMLR, 2018.
> > >
> > > Ruis, L., et al. "A Benchmark for Systematic Generalization in Grounded Language Understanding." Advances in Neural Information Processing Systems 33 (NeurIPS 2020) (2020).
> > >
> > > Baptista, Ricardo, and Matthias Poloczek. "Bayesian optimization of combinatorial structures." International Conference on Machine Learning. PMLR, 2018.

---

### Decision · Program_Chairs · 2022-01-20

**Decision:**

Reject

**Comment:**

The Authors study the emergence of systematic generalization in neural networks. The paper studies a timely topic and presents a set of concrete results. For example, reviewer ZgRW emphasizes that a key strength of the paper is constructing simple datasets where systematicity emerges. I think indeed it is valuable, as systematicity is sometimes poorly defined and understood, so building a theoretical testbed might be very helpful.

However, the reviewers found important issues, which the rebuttal was unable to address. Perhaps the key issue (raised e.g. by reviewer 9QCY) is that results do not clearly generalize to more practically relevant settings. What is somewhat missing is a clear set of guidelines or implications for how to improve systematicity in more practically relevant neural networks.

Based on this and other issues raised by the reviewers, unfortunately, I have to recommend rejecting the paper. Thank you for your submission, and I hope that the review process will help you improve the work.